# SAS-Bench: A Fine-Grained Benchmark for Evaluating Short Answer Scoring with Large Language Models

## Abstract

Short Answer Scoring (SAS) is a critical task in automated subjective answer grading, playing an essential role in education, standardized testing, and large-scale assessment systems. However, existing approaches often produce coarse-grained scores and lack detailed reasoning. Although large language models (LLMs) have demonstrated potential as zero-shot evaluators, they remain susceptible to bias, inconsistencies with human judgment, and limited transparency in scoring decisions. To overcome these limitations, we introduce SAS-Bench, a benchmark specifically designed for LLM-based SAS tasks. SAS-Bench provides fine-grained, step-wise scoring, expert-annotated error categories, and a diverse range of question types derived from real-world subject-specific exams. This benchmark facilitates detailed evaluation of model reasoning processes and explainability. We also release an open-source dataset containing 1,030 questions and 4,109 student responses, each annotated by domain experts. Furthermore, we conduct comprehensive experiments with various LLMs, identifying major challenges in scoring science-related questions and highlighting the effectiveness of few-shot prompting in improving scoring accuracy. Our work offers valuable insights into the development of more robust, fair, and educationally meaningful LLM-based evaluation systems.

## 1 Introduction

Short Answer Scoring (SAS) (Wu & Yeh, 2019; Wu & Shih, 2018; Menini et al., 2019) is a critical, close-ended Subjective Answer Grading (SAG) task (Das et al., 2022; Clay, 2001; Boyd, 1989). It is typically formulated as a pointwise evaluation problem, where student responses, constructed in their own words, are assessed against a reference answer. The significance of SAS lies in its vital role across education, standardized testing, and automated assessment systems, particularly for evaluating short-answer or proof-based questions where constructed responses are essential. Traditional approaches to SAS, however, have primarily focused on producing a single overall score for each response, often lacking fine-grained analysis. With recent advancements in large language models (LLMs) (OpenAI, 2023; DeepSeek-AI, 2024), novel paradigms have emerged in which LLMs serve as judges for zero-shot subjective scoring. These methods enable more comprehensive and nuanced assessments by incorporating multiple evaluation dimensions.

Despite the advantages of using LLM-as-a-judge in zero-shot evaluation tasks, several challenges remain to be addressed. First, current LLMs exhibit notable biases in evaluation tasks, especially when compared to human assessors. These models are influenced by factors such as the positioning of key phrases within the text and the overall length of the content (Raina et al., 2024), leading to biased scoring outcomes. Besides, LLMs tend to exhibit evaluation biases depending on the type of scoring metric used, whether it is a binary assessment (e.g., "yes" or "no") or a numerical scale (e.g., a 1–5 rating) (Zhuang et al., 2024). Consequently, without complex prompt engineering or task-specific fine-tuning, the discrepancies between model-generated scores and human evaluations increase significantly, and the performance of LLMs in this task can even fall below that of smaller language models (SLMs) (See Appendix J). Second, existing methods lack the capability to provide accurate and interpretable explanations for their evaluation results (Deshpande et al., 2024). For instance, these models fail to offer clear justifications for specific score assignments or to pinpoint the

errors that contributed to inaccurate assessments, making manual verification more difficult. These limitations significantly hinder the practical deployment of LLM-based evaluation systems. Moreover, previous studies on SAS and related Automated Essay Scoring (AES) tasks (Chamieh et al., 2024; Kim & Jo, 2024; Lee et al., 2024) primarily evaluate overall scores and rarely examine model behavior through bias analysis or other diagnostic perspectives. A major reason is that existing datasets typically contain only questions, reference answers, human-annotated scores, and categorical labels, which restricts the ability to conduct detailed analyses of scoring behavior. Motivated by this limitation, our work introduces step-wise scores and error-cause annotations that support more fine-grained evaluation of LLM scoring performance and enable more systematic investigation of potential model biases.

To better understand how these challenges impact the SAS task, a benchmark specifically designed for generative language models is essential. As illustrated in Figure 1, most existing benchmarks (Barbara et al., 2012; Mohler & Mihalcea, 2009; Lai et al., 2024) rely on a simple input structure consisting of the question, reference answer, and student response, and evaluate model performance based on the alignment between predicted overall scores and human annotations. While this approach provides a general sense of scoring accuracy, it fails to capture the model's reasoning process and its ability to evaluate diverse forms of student responses. In particular, existing benchmarks still exhibit the following limitations: (1) Lack of Fine-Grained Evaluation. Existing automatic scoring benchmarks focus on measuring the overall consistency between model-generated and human-assigned scores. However, they fail to capture how different response formats or key phrases within student answers may affect the model's scoring bias. Such fine-grained evaluation is particularly important for long-form responses, especially in science subjects, where accurate scoring often depends on specific reasoning steps and domain-specific terminology. (2) Limited Explainability Assessment. While automatic scoring models are bound to make occasional errors, their ability to interpret and justify their predictions is essential for enabling effective human oversight and review. Unfortunately, current benchmarks lack robust quantitative methods to assess the model's explainability. This shortcoming significantly hinders the adoption of these systems in high-stakes educational or assessment contexts.

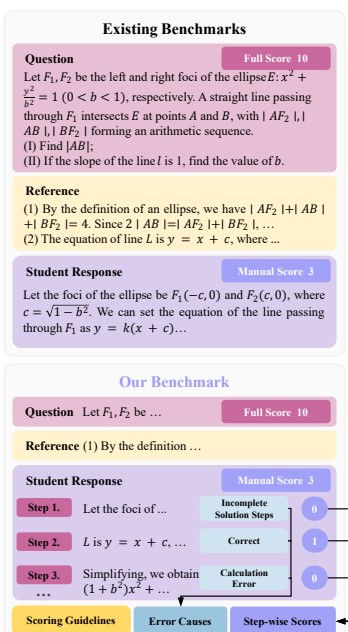

Figure 1: Comparison of existing benchmarks and our benchmark.

To address these limitations, we present SAS-Bench, a new benchmark specifically designed for the SAS task. Constructed from authentic questions in China's National College Entrance Examination (Gaokao), SAS-Bench covers nine academic subjects with 1,030 questions and 4,109 student responses, collected through a collaborative strategy that combines real student responses with LLM-synthesized data. All responses are manually annotated by subject-matter experts with step-wise scores and detailed error causes to ensure the accuracy and reliability of evaluation results. The benchmark also incorporates rigorous evaluation protocols to support consistent and fair comparison across models. ***To support fine-grained assessment of model scoring performance***, the dataset includes multiple template-free question types. Each student's response is segmented into multiple steps based on key phrases, allowing for detailed evaluation of how models handle different answer structures and reasoning processes. ***To quantify model explainability***, SAS-Bench incorporates a predefined set of error causes for each question type. Every step of each student's response is paired with expert-annotated error labels, enabling a systematic comparison between model-predicted and human-identified error causes. This facilitates a robust assessment of the model's ability to explain its scoring decisions. To evaluate the current capabilities of LLMs in SAS tasks, we conduct comprehensive experiments on sixteen widely used LLMs. Our results highlight that step-wise scoring and error causes inference remain particularly challenging in SAS tasks. Additionally, we find that providing few-shot demonstrations and scoring guidelines is positively correlated with improved model performance, offering valuable insights for enhancing LLM-as-a-Judge systems. In summary, our key contributions are as follows:

- *New Benchmark.* We present SAS-Bench, the first benchmark specifically tailored for SAS with LLMs. It features step-wise scoring and a predefined taxonomy of error causes, enabling more comprehensive evaluation of models' reasoning processes and the interpretability of their scoring decisions.

- *Accessible Dataset.* We construct and publicly release an open-source dataset comprising 1,030 authentic questions with 4,109 student responses, each meticulously annotated by subject-matter experts with step-wise scores and detailed error labels, ensuring accurate and reliable evaluation.

- *Comprehensive Evaluation.* We conduct extensive experiments on sixteen widely adopted LLMs, identifying major challenges in scoring short-answer responses and offering practical insights for enhancing LLM-as-a-Judge systems.

## 2 RELATED WORK

### 2.1 AUTOMATED SCORING BENCHMARKS

Current automated scoring benchmarks can generally be categorized into two types: (1) Open-ended AES benchmarks without reference answers, such as the Kaggle ASAP-AES (Hamner et al., 2012), ScAA (Agarwal et al., 2020), SciEntsBank (Dzikovska et al., 2013), and the more recent RiceChem (Sonkar et al., 2024), which is tailored for evaluating LLMs. These datasets typically include question prompts, student essays, expert-assigned scores, and auxiliary metadata such as topic or user information. The recently introduced ENEM dataset (Silveira et al., 2024) further enriches evaluation by incorporating expert commentary on dimensions such as topic relevance and linguistic quality. (2) Close-ended SAS benchmarks with reference answers, which typically include a question prompt, a student response, and a corresponding reference answer. Representative datasets include the Kaggle ASAP-SAS (Barbara et al., 2012), which spans multiple subjects including science, mathematics, and language arts; the SR dataset (Menini et al., 2019), which focuses on medical-domain questions; the ASAG dataset (Mohler & Mihalcea, 2009), targeting short-answer questions in computer science; and the LE dataset (Lai et al., 2024), which contains short-answer responses from the field of logistics management. The recently proposed L-Eval benchmark (An et al., 2024) encompasses both open-ended and close-ended tasks, and introduces improved evaluation metrics for evaluating LLM grading performance on long-form responses. Despite their contributions, most existing datasets include only raw student responses and do not account for the influence of step-wise responses.

### 2.2 LLM-AS-A-JUDGE SYSTEMS

With the rapid advancement of instruction-following capabilities in LLMs, the LLM-as-a-Judge paradigm has attracted growing interest. Numerous studies have explored prompting LLMs to directly assign assessment scores (Wang et al., 2023; Thomas et al., 2024; Faggioli et al., 2023; Kazi & Kahanda, 2023). However, as this line of research progresses, several limitations have been identified. Prior work on SAS and related Automated Essay Scoring (AES) tasks has consistently shown that LLM performance in zero-shot and few-shot settings remains inadequate, as reported in Chamieh et al. (2024); Kim & Jo (2024); Lee et al. (2024). Notably, LLMs exhibit high sensitivity to superficial variations in student responses, such as changes in the position or length of key phrases, which can significantly influence scoring behavior and make them vulnerable to adversarial attacks (Raina et al., 2024). In addition, LLMs often produce scoring outcomes that are inconsistent with human judgments, largely due to their susceptibility to variations in scoring rubrics or prompt wording (Zhuang et al., 2024). To mitigate these challenges, recent work has proposed aligning LLM scoring with human evaluation standards by incorporating reference answers (Zheng et al., 2023; Zhu et al., 2023), detailed scoring rubrics (Liu et al., 2024; Stahl et al., 2024; Latif et al., 2024), and multi-perspective evaluation frameworks (Liusie et al., 2024). These insights have informed the design of our benchmark, in which we provide explicit scoring guidelines and carefully curated reference answers to support a more rigorous, fair, and interpretable evaluation of LLM-based scoring systems.

## 3  SAS-BENCH

The overall workflow of SAS-Bench is illustrated in Figure 2. Section 3.1 presents the underlying principles guiding our data design. Section 3.2 details the dataset construction pipeline along with key statistics. Finally, Section 3.3 introduces the evaluation methodology and defines the metrics used in our benchmark.

### 3.1  BENCHMARK DESIGN

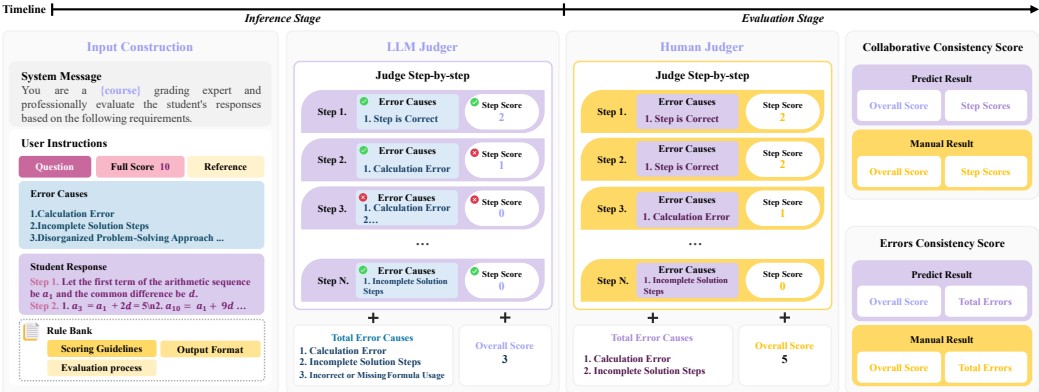

Figure 2: The workflow of our SAS-Bench. The results from the human judger are predefined during the dataset construction.

The motivation behind our benchmark design is driven by three key questions: 1) Can the LLM accurately assign overall scores across varying levels of student performance? 2) Can the LLM reliably evaluate step-wise scores for multi-step responses? 3) Can the LLM effectively identify and analyze the underlying causes of errors in student responses? To address these questions, we propose a LLM-as-a-Judge framework, in which the model is prompted via a system message to act as a subject-matter teacher. During the inference stage, each input instance includes the question, the full score, the corresponding reference answer, the associated set of error causes, the student's response, and a rule bank (containing scoring guidelines, output formats, etc.). Given this input, the model is tasked with generating the overall score, step-wise scores, and step-wise error causes for each response. In the evaluation stage, we assess the model's performance on the SAS task by comparing its judgments with human annotations across these three dimensions:

**Overall Score Consistency.** Overall score consistency is a core metric in the SAS task, as it reflects the model's ability to provide accurate holistic evaluations of student responses. By comparing model-predicted scores with human-assigned scores, we can assess how well the model aligns with expert judgment. The reliability of this metric directly impacts the credibility and utility of the automated scoring system.

**Step-wise Score Consistency.** While many existing SAS methods focus primarily on predicting overall scores, they often lack finer-grained evaluations. In practice, model-generated scores may not always be persuasive, particularly in complex multi-step problems. Evaluating the consistency between model-predicted and human-assigned step-wise scores offers a more detailed assessment of the model's scoring logic. Additionally, science subjects often involve multi-step reasoning, where the outcome of one step may influence subsequent steps. Step-wise score consistency thus serves as an indirect measure of the model's ability to capture inter-step dependencies and perform causal reasoning.

**Error Cause Consistency.** Explainability is a critical concern in SAS and related LLM-as-a-Judge tasks (Deshpande et al., 2024; Stahl et al., 2024). To address this, we propose explicitly predicting the underlying causes of errors in student responses. This not only enhances transparency in model decision-making but also allows us to assess the alignment between model interpretations and human judgments. Measuring the consistency of error cause predictions provides valuable insight into the model's interpretability and its potential to offer actionable feedback.

## 3.2 DATASET

Following the proposed design framework, the SAS-Bench dataset is developed through a structured pipeline comprising data collection, data cleaning, error cause set construction, and manual annotation. This process results in a high-quality dataset that incorporates multi-dimensional evaluation metrics.

**Data Collection.** Our dataset targets the high school level, aiming to cover challenging evaluation scenarios across diverse subjects. We adopt the publicly available GAOKAO-Benchmark (Zhang et al., 2023), based on China's National College Entrance Examination (Gaokao), as our source of questions. Most items in the dataset are in Chinese, except for English subject. To evaluate LLM-as-a-Judge under varied score distributions and potential position and length biases, we designed a collaborative strategy combining real student responses with LLM-synthesized data. Beyond collecting answers to traditional short-answer questions, we convert multiple-choice questions (MCQs) and gap-filling questions into short-answer formats using a "template-free" approach (See Appendix L), ensuring a more uniform score distribution and introducing the necessary biases for robustness testing. We recruited three science and three humanities students to provide authentic responses reflecting their subject strengths. Since LLMs are more efficient at producing both perfect responses and responses with significant errors, we augmented the dataset with a 1:1 ratio of positive to negative samples. Positive samples were generated by prompting LLMs to rewrite reference answers in varied styles or directly answer questions. Negative samples were constructed as follows: (1) for template-free MCQs, we analyzed the structure and correct options, randomly selected distractors via rules, and prompted LLMs to express them freely; (2) for gap-filling and short-answer items, we sampled fragments from reference answers based on target score distributions and instructed the LLM to introduce errors at specific points; (3) for short-answer questions, we decomposed reference answers into logical steps, then randomly removed some steps, and used a smaller LLM to fill in the blanks, producing more natural incorrect responses.

**Data Cleaning.** For each question, we collect eight total samples. During the data cleaning phase, we focus on removing overly similar responses. Initially, we use GPT-4o to score the answers and group them based on the resulting score distribution. We then compute BLEU and Jaccard similarity scores between pairs of answers within each group, filtering out those with high redundancy. This process ensures that the final dataset preserves representative examples across different score levels while effectively eliminating synonymous paraphrases and heavily overlapping samples, thereby mitigating issues related to distribution imbalance.

**Error Causes Construction.** To evaluate the explainability of LLM judges, it is essential to assess their ability to identify and analyze the underlying causes of errors. However, quantitatively measuring this explainability is challenging, largely due to the diversity of unstructured responses and the difficulty of converting free-form explanations into a standardized numerical format. To address this challenge, we propose constructing a structured list of potential error causes for each subject and question type. By constraining LLMs to select error causes from this predefined list when explaining their scoring decisions, we enable a more systematic and quantifiable evaluation of model explainability. Specifically, for each subject and question type, we adopt a few-shot prompting strategy using representative questions, reference answers, and filtered responses as input to GPT-4o, prompting it to generate 50 sets of error cause descriptions. Human annotators then consolidate and refine these descriptions, merging similar entries and simplifying the language to produce a comprehensive set of error causes that captures all major categories of mistakes.

**Human Annotation.** We recruited 18 subject-matter experts and divided them evenly into two groups: one for initial annotation and the other for verification. The first group annotated the data, while the second group reviewed and validated these annotations. In cases of significant discrepancies between the two groups' evaluations, the annotators engaged in discussions to reach a consensus, followed by re-annotation if necessary. During the annotation process, annotators were instructed to: (1) segment each response into distinct steps (see the step division method in Appendix O), (2) assign a score to each step, (3) label the error cause for each step, and (4) provide an overall score for the response. After completing the annotations for each subject and question type, annotators compiled a corresponding scoring guideline to support consistency in subsequent model-based evaluations.

Table 1: Statistics of our dataset. "Avg. Steps" and "Max Steps" denote the average and maximum number of steps per response, respectively. "Avg. Length" denotes the average length of per response.

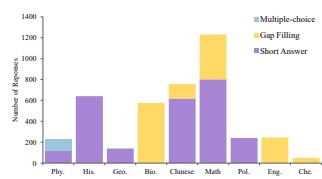

| Question Type | Phy. | His. | Geo. | Bio. | Chi. | Math | Pol. | Eng. | Che. | Total |
|---|---|---|---|---|---|---|---|---|---|---|
| Questions | 47 | 128 | 28 | 116 | 181 | 412 | 60 | 49 | 9 | 1,030 |
| Error Causes | 10 | 5 | 5 | 7 | 9 | 9 | 4 | 9 | 7 | 65 |
| Avg. Steps | 4.9 | 1.8 | 3.3 | 2.8 | 2.8 | 5.4 | 4.6 | 9.6 | 5.1 | - |
| Max Steps | 16 | 8 | 10 | 8 | 12 | 25 | 12 | 16 | 8 | - |
| Responses | 230 | 640 | 140 | 576 | 757 | 1,231 | 240 | 245 | 50 | 4,109 |
| Avg. Length | 867.7 | 303.9 | 322.4 | 83.1 | 181.7 | 831.3 | 299.4 | 167.0 | 522.4 | - |

Figure 3: Distribution of question types across different subjects.

**Data Statistics.** As shown in Table 1, SAS-Bench comprises 4,109 responses across nine subjects: Physics, History, Geography, Biology, Chinese, Mathematics, Politics, English, and Chemistry, with a response-to-question ratio of about 4:1. Each response contains an average of six annotated error causes. Science subjects generally involve more detailed step-wise reasoning than humanities, with Mathematics reaching up to 25 steps in a single response. Figure 3 presents the distribution of question types. The dataset is primarily composed of short answer and gap filling questions. In addition to MCQs in Physics, the Chinese subject includes 440 responses that embed template-free MCQs within short answer questions. We further analyze the complexity of different question types in Appendix N.

### 3.3 Evaluation Protocol

Most existing benchmarks (Meyer et al., 2024; Mohler et al., 2011; Dzikovska et al., 2013) provide only manually annotated overall scores, enabling model evaluation solely based on final scoring outcomes. However, they fall short in assessing step-wise scoring accuracy and the ability to identify specific error causes within responses. To more comprehensively evaluate LLM performance on this task, our benchmark simultaneously considers the alignment between model predictions and human annotations across three dimensions: overall scores, step-wise scores, and error cause interpretation.

While previous studies commonly adopt metrics such as Quadratic Weighted Kappa (QWK) (Yoon, 2023; Khayi & Rus, 2024), Pearson correlation (Deshpande et al., 2024; Bonthu et al., 2023), or Spearman correlation (Liu et al., 2023; Chuang et al., 2022) to assess scoring consistency, these metrics are primarily designed for evaluating overall scores and are not directly applicable to the multi-dimensional evaluation framework we propose. To overcome this limitation, we extend these traditional metrics and introduce two new evaluation measures tailored:

**Collaborative Consistency Score (CCS).** To jointly evaluate the consistency between model predictions and human annotations at both the overall and step-wise scoring levels, we propose the Collaborative Consistency Score (CCS). Formally, let $r_i$ and $r_j$ denote the holistic scores (e.g., overall scores) assigned by the model and human annotators, respectively, and let $s_{i,k}$ and $s_{j,k}$ denote their corresponding step-specific scores for the $k$-th step ($k \in [1, m]$). Then we define an adjusted weight matrix $W$ that captures both holistic and step-wise discrepancies:

$$W_{i,j} = \alpha \cdot \frac{(r_i - r_j)^2}{(N_r - 1)^2} + \frac{1 - \alpha}{m} \sum_{k=1}^{m} \frac{(s_{i,k} - s_{j,k})^2}{(N_{s_k} - 1)^2}, \tag{1}$$

where $\alpha \in [0, 1]$ controls the trade-off between holistic and step-wise differences (e.g., $\alpha = 0.5$), $N_r$ denotes the number of possible holistic score levels, $N_{s_k}$ denotes the number of possible score levels for step $k$, and $m$ is the total number of steps. The CCS is then computed as:

$$\text{CCS} := 1 - \frac{\sum_{i,j} O_{i,j} \cdot W_{i,j}}{\sum_{i,j} E_{i,j} \cdot W_{i,j}}, \tag{2}$$

where $O$ represents the observed matrix and $E$ represents the expected matrix, both following the standard definitions used in the QWK metric.

**Errors Consistency Score (ECS).** To evaluate an LLM's ability to interpret error causes, we propose the Errors Consistency Score (ECS), which measures the consistency between model and human annotations in terms of error distributions. Although error causes are annotated at each step,

Table 2: CCS score (%) comparison across various LLMs, evaluated by subject area and question type. **S.** indicates Short Answer, **M.** denotes Multiple Choice, and **G.** refers to Gap Filling. The best and second-best scores in each category are highlighted in **bold** and underlined, respectively. Standard deviations are provided in parentheses.

| Models | Phy. (S.) | Phy. (M.) | His. (S.) | Geo. (S.) | Bio. (G.) | Chi. (G.) | Chi. (S.) | Math (S.) | Math (G.) | Pol. (S.) | Eng. (G.) | Che. (G.) | Avg. |
|---|---|---|---|---|---|---|---|---|---|---|---|---|---|
| *Reasoning-based LLMs* | | | | | | | | | | | | | |
| Deepseek-R1 | 38.43 (4.12) | **95.01 (0.63)** | **80.98 (7.02)** | 67.92 (2.48) | **79.12 (1.08)** | 95.09 (0.46) | 69.07 (4.91) | 57.85 (2.01) | **83.56 (9.81)** | 71.92 (5.21) | 73.19 (8.44) | 72.92 (3.48) | 73.76 (2.33) |
| QwQ-32B | 48.53 (4.33) | 87.23 (0.71) | 75.43 (6.11) | **77.06 (2.12)** | 72.52 (1.22) | **96.00 (0.57)** | 31.77 (4.38) | 48.66 (2.35) | 45.51 (9.12) | 74.48 (5.14) | 54.79 (8.76) | 62.17 (3.57) | 64.51 (2.41) |
| TinyR1-32B-Preview | 38.17 (3.74) | 84.88 (0.69) | 75.83 (6.71) | 71.52 (2.74) | 73.45 (0.89) | 92.57 (0.33) | 52.61 (5.38) | 48.28 (2.14) | 44.77 (9.57) | 70.70 (4.96) | 57.92 (8.33) | 41.37 (3.12) | 65.17 (2.29) |
| Qwen3-32B | 47.29 (4.58) | 85.51 (0.63) | 64.96 (7.44) | 80.43 (2.61) | 63.15 (1.05) | 92.21 (0.48) | 50.43 (5.11) | 51.26 (2.06) | 80.77 (8.94) | 73.30 (5.38) | 59.33 (7.91) | 57.82 (3.51) | 67.20 (2.63) |
| Qwen3-8B | 54.33 (5.18) | 76.17 (0.83) | 45.54 (7.92) | 68.89 (2.41) | 43.22 (1.27) | 86.01 (0.66) | 42.02 (5.47) | 46.33 (2.57) | 73.33 (10.18) | 64.25 (6.74) | 50.55 (9.88) | 50.52 (4.42) | 58.43 (2.84) |
| MiMo-7B-RL | 52.77 (5.02) | 41.01 (0.64) | 61.33 (8.44) | 67.10 (2.62) | 35.93 (1.35) | 54.72 (0.71) | 43.09 (5.16) | 38.09 (2.74) | 55.79 (12.01) | 36.78 (6.02) | 34.69 (9.44) | 31.05 (4.33) | 46.03 (2.74) |
| Deepseek-Prover-V2-7B | 22.59 (4.66) | 10.75 (0.83) | 2.92 (7.31) | 30.71 (2.58) | 50.63 (1.44) | 55.48 (0.62) | 12.95 (6.18) | 0.87 (2.92) | 2.29 (12.12) | 10.44 (6.44) | 30.19 (9.02) | 28.76 (3.88) | 21.55 (2.88) |
| DeepSeek-R1-Distill-7B | 33.71 (5.14) | 29.24 (0.92) | 50.92 (8.02) | 32.35 (2.12) | 52.18 (1.31) | 52.44 (0.77) | 44.29 (6.05) | 29.52 (2.63) | 39.55 (11.37) | 53.77 (5.91) | 32.98 (9.55) | 34.27 (4.51) | 40.44 (2.96) |
| *RLHF-based LLMs* | | | | | | | | | | | | | |
| Deepseek-V3 | 53.89 (3.71) | 85.72 (0.73) | 69.85 (6.24) | 76.23 (2.18) | 76.51 (1.12) | 93.42 (0.44) | **69.49 (4.52)** | **58.81 (1.94)** | 80.18 (8.98) | **76.75 (4.68)** | **73.82 (7.52)** | **74.64 (3.11)** | **74.11 (2.02)** |
| GPT 4o-mini-20240718 | **58.90 (4.77)** | 81.19 (0.71) | 54.85 (7.62) | 76.59 (2.67) | 65.39 (1.27) | 87.65 (0.51) | 55.25 (5.12) | 43.56 (2.25) | 37.38 (10.54) | 63.44 (5.63) | 22.60 (9.42) | 55.98 (3.81) | 58.56 (2.48) |
| Llama3.3-70B-Instruct | 45.34 (3.89) | 70.03 (0.62) | 72.02 (6.74) | 72.51 (2.41) | 67.94 (1.02) | 85.30 (0.43) | 35.83 (4.67) | 58.60 (1.82) | 74.97 (9.51) | 63.68 (4.81) | 67.60 (7.88) | 38.94 (3.22) | 62.73 (2.21) |
| Mixtral 8×7B-Instruct | 30.78 (4.44) | 42.27 (0.83) | 33.43 (7.88) | 4.99 (2.77) | 44.45 (1.55) | 29.85 (0.74) | 24.00 (6.04) | 26.73 (2.85) | 70.04 (11.41) | 43.92 (6.58) | 33.40 (10.33) | 42.05 (4.42) | 35.49 (2.71) |
| Qwen2.5-32B-Instruct | 40.53 (4.54) | 77.02 (0.67) | 62.34 (6.98) | 74.50 (2.54) | 72.07 (1.21) | 94.85 (0.59) | 66.37 (4.74) | 50.08 (2.33) | 32.59 (10.52) | 64.09 (5.37) | 53.35 (8.55) | 62.87 (3.44) | 62.56 (2.52) |
| Qwen2.5-14B-Instruct | 53.76 (4.22) | 66.12 (0.74) | 60.96 (7.48) | 74.30 (2.88) | 67.50 (1.33) | 92.81 (0.63) | 63.08 (5.12) | 43.28 (2.11) | 75.62 (9.88) | 62.03 (6.01) | 56.34 (9.11) | 57.53 (3.56) | 64.44 (2.39) |
| GLM4-9B-Chat | 45.62 (5.06) | 52.33 (0.91) | 36.81 (7.12) | 69.41 (2.53) | 39.19 (1.41) | 63.92 (0.71) | 42.94 (5.61) | 35.50 (2.74) | 56.95 (11.12) | 54.83 (6.11) | 33.92 (10.21) | 30.79 (4.55) | 46.85 (2.84) |
| Llama3-8B-Instruct | 41.09 (4.28) | 35.10 (0.78) | 37.52 (8.34) | 31.29 (2.74) | 32.19 (1.47) | 38.13 (0.67) | 32.89 (6.47) | 23.55 (2.88) | 62.43 (12.24) | 37.78 (6.88) | 31.68 (9.77) | 29.27 (4.28) | 36.08 (2.74) |

we observe that both the number of steps and error types are relatively sparse, leading to instability in fine-grained step-wise consistency evaluation. Therefore, we aggregate error causes across all steps within a response, preserving both their types and frequencies. To further enhance stability, we partition the evaluation samples into score intervals based on their normalized overall scores and compute consistency scores within each interval. Specifically, let $p$ and $g$ denote the normalized overall scores predicted by the model and assigned by human annotators, respectively. Let $\mathbf{E}^p, \mathbf{E}^g \in \mathbb{R}^{n \times l}$ represent the error frequency matrices for the model and human annotations, where $n$ is the number of samples and $l$ is the number of error types. Given $m$ quantile thresholds $\{\tau_q\}_{q=1}^{m-1}$ that divide the samples into $m$ score intervals, we define the interval assignment function and the corresponding accumulated error frequencies as follows:

$$\phi(x) = \sum_{q=1}^{m-1} \mathbb{I}(x \geq \tau_q), \quad \mathbf{M}_k^p[j] = \sum_{i:\phi(p_i)=k} \mathbf{E}_{i,j}^p, \quad \mathbf{M}_k^g[j] = \sum_{i:\phi(g_i)=k} \mathbf{E}_{i,j}^g, \qquad (3)$$

where $\mathbb{I}(\cdot)$ is the indicator function, $k \in \{0, 1, \ldots, m-1\}$ denotes the interval index, and $j \in [1, l]$ indexes the error types. We then compute the Spearman rank correlation coefficient $\rho_k$ between $\mathbf{M}_k^p$ and $\mathbf{M}_k^g$ within each interval, and the ECS is computed as the average correlation across all intervals:

$$\text{ECS} := \frac{1}{m} \sum_{k=0}^{m-1} \rho_k, \quad \rho_k = \text{SpearmanR}(\mathbf{M}_k^p, \mathbf{M}_k^g). \qquad (4)$$

## 4 EXPERIMENTS

### 4.1 EXPERIMENTAL SETUP

In the experiments, we evaluate SAS-Bench using LLMs of varying scales and types. Our analysis centers on three core challenges in applying LLMs to SAS tasks: (1) the consistency of overall score predictions across student responses of different quality levels; (2) the alignment between overall scores and step-wise evaluations in multi-step responses; and (3) the reliability of LLMs in predicting the frequency of different error causes in student responses.

**Metrics.** We adopt the CCS and ECS metrics introduced in § 3.3 to evaluate, respectively, the consistency between overall and step-wise scores and the consistency of model predictions on the frequency of different error causes. Additionally, we use Quadratic Weighted Kappa (QWK) to assess overall scoring performance at a high level, and employ standard metrics such as the F1 score to support performance analysis in specific evaluation scenarios.

**Models.** We conduct experiments on 16 LLMs, which are broadly classified into two categories: RLHF-based models and reasoning-based models. The RLHF-based models include Deepseek-V3 (DeepSeek-AI, 2024), GPT-4 (OpenAI, 2023), the LLaMA family (Dubey et al., 2024), Qwen-2.5 (Yang et al., 2024a;b), GLM-4 (GLM et al., 2024), Mixtral (Jiang et al., 2024), and Deepseek-Prover-V2 (Ren et al., 2025). The reasoning-based models comprise Deepseek-R1 (DeepSeek-AI et al., 2025), QwQ (Team, 2025b), Qwen-3 (Team, 2025a), Tiny-R1 (Team, 2025c), and MiMo-RL

(Xiaomi LLM-Core Team, 2025). Among these, LLaMA3-MetaMath (Yu et al., 2024), MiMo-RL, Deepseek-Prover-V2, and Tiny-R1 are proprietary models specifically optimized for mathematical problem solving.

**Settings.** In the data synthesis process described in § 3.2, we use GPT-4o-mini and Deepseek-V3 as large-scale LLMs to generate synthetic data. For smaller-scale models, including GLM4-9B, Qwen2.5-14B, and LLaMA3-8B, we use them to complete predefined incomplete templates. To increase the diversity of the generated content, we introduce randomness by varying the temperature during the generation process. For evaluation, we evaluate all models on $4\times$ NVIDIA A800 80G, and set the temperature of all reasoning-based models to 0.6 to optimize their reasoning capabilities. For RLHF-based models, we default to a temperature of 0.0. However, if a model produces outputs with invalid formatting, we re-run the generation at a temperature of 0.6 to obtain valid responses.

### 4.2 SCORING CONSISTENCY RESULTS

We evaluate the consistency between overall and step-wise scores using the CCS metric, with results across different subjects and question types presented in Table 2. To highlight the distinction between CCS and traditional overall score consistency metrics, we additionally report each model's QWK performance in the radar charts of Figure 4. We also present the distribution of model-predicted overall scores in Appendix I.

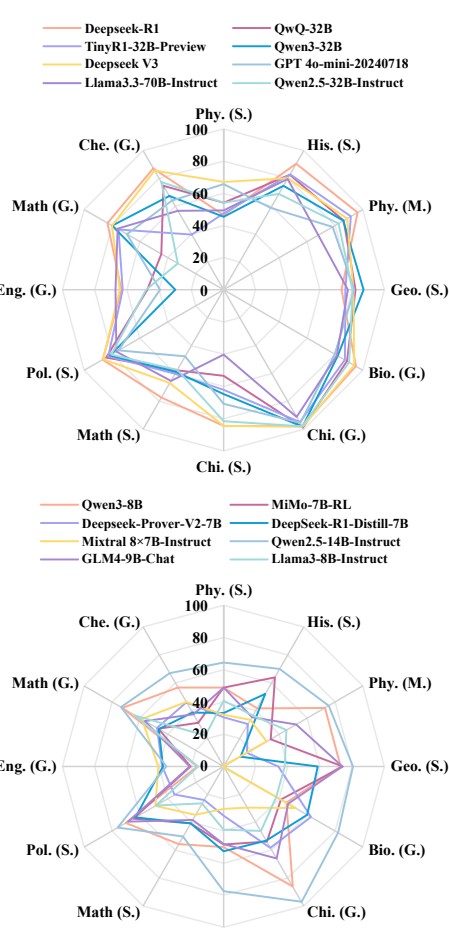

Figure 4: Comparison of QWK scores across LLMs.

Overall, we observe a positive correlation between CCS and QWK scores across all models, as well as between model size and average performance. Notably, Deepseek-V3 and Deepseek-R1 achieve the best average results in CCS and QWK metrics, respectively. A comparison of Table 2 and Figure 4 shows that CCS scores are generally lower than QWK scores, indicating that incorporating step-wise consistency introduces additional challenges for scoring models. Models tend to perform better on humanities-related questions in both metrics, while science-related questions remain more difficult. For larger models with over 32B parameters, such as LLaMA3.3-70B-Instruct and Qwen3-32B, the CCS scores on short-answer questions like **His. (S.)** and **Chi. (S.)** are notably lower than their corresponding QWK scores. This discrepancy is more pronounced in reasoning-based models than in RLHF-based models. For instance, TinyR1-32B exhibits an 11.89% drop in CCS compared to QWK on **Math (S.)**, suggesting that many LLMs still struggle with step-wise scoring consistency. Conversely, there are also counterexamples. MiMo-7B-RL, for example, achieves CCS scores on **Phy. (M.)**, **Phy. (S.)**, and **Eng. (G.)** that are, on average, 11.47% higher than its QWK scores. This demonstrates that CCS provides a valuable and complementary perspective for evaluating model performance in SAS tasks. Finally, smaller-scale models (with fewer than 32B parameters) exhibit noticeable weaknesses in gap-filling tasks (**Eng. (G.)**) and template-free MCQs (**Phy. (M.)**), suggesting limited ability to handle sparse semantic content.

Building on the above experimental findings, the CCS metric offers more fine-grained insights into model performance on SAS tasks. By introducing step-wise consistency evaluation, CCS can guide fu-

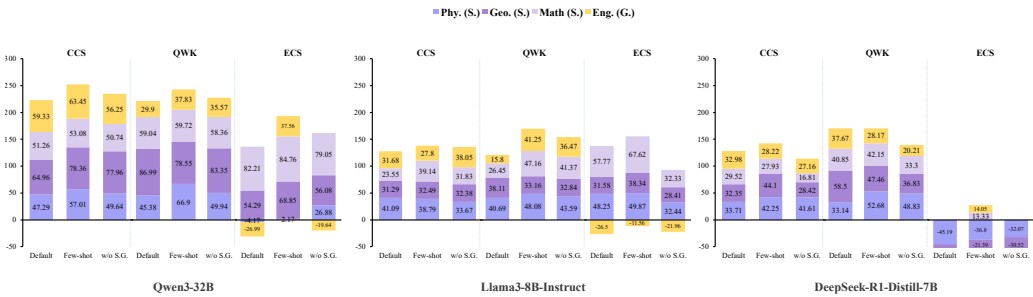

Figure 5: Performance changes of selected models across various subjects under settings of "few-shot" and scenarios without scoring guidelines ("w/o S.G.").

ture work toward enhancing step-by-step reasoning,
thereby improving the practical utility of models in real-world scenarios.

## 4.3 ERRORS CONSISTENCY RESULTS

Table 3: ECS score (%) comparison across various LLMs, evaluated by subject area and question type. **S.** indicates Short Answer, **M.** denotes Multiple Choice, and **G.** refers to Gap Filling. The best and second-best scores in each category are highlighted in **bold** and underlined, respectively. Standard deviations are provided in parentheses.

| Models | Phy. (S.) | Phy. (M.) | His. (S.) | Geo. (S.) | Bio. (G.) | Chi. (G.) | Chi. (S.) | Math (S.) | Math (G.) | Pol. (S.) | Eng. (G.) | Che. (G.) | Avg. |
|---|---|---|---|---|---|---|---|---|---|---|---|---|---|
| | | | | | | Reasoning-based LLMs | | | | | | | |
| Deepseek-R1 | 23.25 (6.91) | 30.59 (12.32) | 57.53 (7.44) | 56.08 (1.67) | 69.20 (1.43) | 86.04 (10.01) | 72.68 (10.12) | **94.29** (9.83) | 15.20 (16.10) | 65.56 (5.92) | 18.65 (8.34) | 81.76 (13.52) | **55.90** (5.12) |
| QwQ-32B | 4.74 (7.32) | **63.92** (13.55) | 67.06 (8.42) | 70.04 (2.11) | 53.68 (1.66) | 51.08 (11.28) | 69.20 (9.34) | 79.05 (13.22) | 16.82 (19.92) | 48.81 (6.54) | -22.53 (8.11) | 48.94 (17.08) | 45.90 (5.62) |
| TinyR1-32B-Preview | 3.10 (7.44) | **63.92** (11.24) | 65.71 (9.89) | **77.02** (2.18) | 56.61 (1.53) | 64.42 (10.83) | 74.83 (9.94) | 82.86 (10.67) | 23.33 (18.80) | 40.17 (6.42) | -31.52 (10.74) | 17.35 (16.23) | 44.82 (5.88) |
| Qwen3-32B | -4.17 (8.65) | 24.18 (13.81) | 69.52 (7.15) | 54.29 (2.08) | 53.67 (1.55) | 52.70 (12.02) | 47.31 (12.66) | 82.21 (14.50) | 18.33 (18.04) | 62.14 (6.86) | -26.99 (7.88) | 36.27 (16.88) | 39.12 (5.21) |
| Qwen3-8B | 23.39 (8.98) | **63.92** (12.72) | 14.29 (10.25) | -4.96 (1.76) | 52.21 (1.61) | 47.75 (10.76) | 34.01 (13.42) | 39.20 (13.55) | -8.14 (19.60) | 57.19 (6.01) | -27.13 (9.66) | 59.28 (14.10) | 29.25 (5.09) |
| MiMo-7B-RL | **51.05** (9.32) | 24.18 (12.10) | 14.29 (9.11) | 38.85 (2.22) | 58.35 (1.48) | 92.17 (5.85) | 63.07 (13.18) | 13.39 (14.79) | 35.12 (20.22) | -27.10 (7.11) | -4.41 (6.23) | 1.04 (16.44) | 30.00 (5.33) |
| Deepseek-Prover-V2-7B | -24.10 (8.44) | -5.20 (13.50) | 42.86 (8.34) | -6.23 (2.14) | 29.54 (1.39) | -80.81 (9.67) | 23.25 (8.58) | 46.67 (12.11) | -1.51 (17.77) | -58.64 (6.42) | -45.23 (9.45) | -21.91 (13.22) | -8.44 (5.48) |
| DeepSeek-R1-Distill-7B | -45.19 (6.03) | 24.18 (14.60) | 0.95 (6.88) | -38.66 (1.92) | 23.55 (1.31) | -20.36 (10.90) | 3.87 (11.22) | -23.81 (13.01) | -13.57 (17.30) | -18.81 (6.12) | -19.59 (10.33) | -44.58 (16.67) | -14.34 (5.72) |
| | | | | | | RLHF-based LLMs | | | | | | | |
| Deepseek-V3 | 7.79 (5.44) | 46.58 (11.44) | 58.10 (6.93) | 32.62 (1.22) | 72.38 (1.10) | **96.58** (8.41) | 57.43 (9.52) | 92.38 (9.93) | 33.33 (12.55) | 40.26 (4.71) | **24.77** (7.82) | **85.83** (10.55) | 54.00 (4.14) |
| GPT 4o-mini-20240718 | 17.91 (6.93) | 24.18 (10.88) | 62.14 (7.12) | 36.68 (1.53) | 55.20 (1.41) | 79.01 (8.88) | **78.00** (8.66) | 67.62 (9.00) | **46.90** (14.99) | **92.31** (5.99) | 10.04 (7.52) | 36.39 (12.77) | 50.53 (4.55) |
| Llama3.3-70B-Instruct | 22.56 (7.32) | 57.35 (12.44) | 54.29 (8.76) | 42.11 (1.62) | 45.09 (1.39) | 52.70 (9.99) | 46.25 (10.41) | 54.29 (10.66) | 30.00 (15.55) | 58.81 (6.83) | -12.53 (9.12) | -15.83 (14.92) | 36.26 (5.44) |
| Mixtral 8x7B-Instruct | 11.99 (7.19) | 17.34 (13.88) | **80.38** (9.44) | 35.84 (1.59) | 32.74 (1.48) | 42.77 (10.45) | 75.82 (10.88) | 56.19 (10.55) | 30.00 (16.33) | 6.84 (6.33) | -31.16 (10.77) | -7.18 (14.41) | 29.30 (5.12) |
| Qwen2.5-32B-Instruct | 11.95 (8.18) | 17.41 (13.33) | 53.33 (7.55) | 59.34 (1.92) | 62.96 (1.66) | 46.90 (11.44) | 75.08 (12.52) | 62.86 (15.68) | 30.00 (19.70) | 46.67 (6.45) | -4.50 (9.22) | 27.08 (16.09) | 40.76 (5.90) |
| Qwen2.5-14B-Instruct | 21.50 (8.41) | 24.18 (14.11) | 47.92 (10.82) | 37.43 (1.73) | **73.36** (1.52) | 64.97 (10.33) | 74.32 (11.03) | 64.94 (13.41) | 18.21 (18.91) | 61.97 (6.23) | -20.00 (10.11) | 47.39 (15.44) | 43.02 (5.66) |
| GLM4-9B-Chat | 35.00 (8.62) | 24.18 (13.74) | 32.49 (9.66) | 34.73 (1.84) | 62.12 (1.57) | 20.36 (10.88) | 77.34 (15.21) | 63.81 (16.44) | **46.90** (19.44) | 82.40 (7.41) | -25.35 (10.34) | 7.18 (16.22) | 38.43 (6.01) |
| Llama3-8B-Instruct | 48.25 (9.18) | 27.46 (14.92) | 17.23 (10.31) | 31.58 (1.66) | 61.37 (1.61) | -14.05 (10.99) | 41.23 (11.91) | 57.77 (12.03) | 21.55 (20.77) | -69.07 (8.37) | -26.50 (11.44) | -27.19 (17.30) | 14.14 (6.81) |

We employ the ECS metric to assess the consistency between model-predicted error cause frequencies and human annotations across different score levels. Specifically, we set $m$ to 3 in Equation 4 to represent low, medium, and high score groups. The performance of LLMs across various subjects and question types is presented in Table 3. We also analyze the relationship between ECS and scoring consistency in Appendix G.

We observe that Deepseek-R1 achieves the highest average ECS score, and overall performance generally improves with increasing model parameter size. However, the results also reveal that current LLMs still face significant challenges in maintaining zero-shot consistency for error causes inference. Additionally, model strengths vary notably across different subjects and question types. The lowest average ECS scores are observed on English gap-filling tasks (**Eng. (G.)**), likely due to the sparse semantics and highly variable answer formats, which make it particularly difficult for LLMs to accurately explain error causes. Compared to scoring tasks, error consistency prediction for short-answer questions in the humanities displays a more polarized pattern: large-scale LLMs achieve relatively high performance, while smaller-scale reasoning-based models tend to make more incorrect inferences. This disparity may be attributed to the lack of explicit alignment between student responses and reference answers in these subjects, which smaller models struggle to resolve due to their limited semantic understanding. Reasoning-based models with fewer than 32B parameters typically perform worse in ECS compared to RLHF-based models. Their more complex reasoning processes often lead to significantly higher inference costs. Overall, ECS effectively highlights the differences between models in terms of consistency in predicting error causes.

## 4.4 IMPACT OF EXAMPLE DEMONSTRATIONS AND SCORING GUIDELINES

To evaluate the impact of example samples and scoring guidelines on model performance, we selected underperforming models and corresponding subject datasets based on Tables 2 and 3. We then examined performance changes under two conditions: providing 5-shot human-scored examples as demonstrations, and removing the scoring guidelines. The results are illustrated in Figure 5. Overall, the few-shot setting leads to improvements in CCS, QWK, and ECS metrics. However, subject-specific analysis reveals that for some subjects like **Phy. (S.)** and **Math (S.)**, few-shot prompting results in decreased CCS scores despite gains in QWK, suggesting that while examples improve overall scoring consistency, the included step-wise labels may mislead the model. Additionally, the absence of scoring guidelines consistently degrades performance across most metrics and tasks, underscoring the importance of providing clear scoring criteria for LLMs in SAS evaluations.

## 5 CONCLUSION

In this study, we introduce SAS-Bench, the first benchmark specifically designed to evaluate LLMs on SAS tasks. Built from real exam data, it includes 1,030 questions and 4,109 expert-annotated responses with step-wise scores and error causes. SAS-Bench introduces fine-grained metrics that assess model performance across overall and step-wise scoring consistency, as well as explainability based on error cause alignment. Our experiments with various LLMs reveal notable performance gaps among LLMs, highlighting SAS-Bench's value in the development of robust and interpretable automated assessment systems.

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

# APPENDIX

## A  LIMITATIONS

Ideally, constructing SAS-Bench entirely from real student responses would yield a distribution that more closely reflects actual application scenarios. However, collecting responses across a full range of scores in a controlled and balanced manner for each subject is difficult in real-world settings. The primary objective of this benchmark is to evaluate the deviations between LLM-based and human assessments in fine-grained response evaluation. To this end, we emphasize the importance of generating diverse response data and engaging professional evaluators for detailed annotations. Looking ahead, we plan to release our annotation system to facilitate contributions from other researchers and support the future expansion of this work.

## B  ETHICS STATEMENT

Our benchmark datasets leverage LLMs to generate responses based on an existing open-source question bank. However, for subjects such as politics and history, the generated content may reflect social biases present in the LLMs' pre-training data. To ensure quality, all annotation experts are practicing high school teachers in their respective subjects, each with over three years of teaching experience, which guarantees that annotations are both reasonable and accurate. Nevertheless, to mitigate the risk of propagating biased information into scoring models, we recommend that any future use of this dataset for training purposes be accompanied by an additional round of manual review.

## C  REPRODUCIBILITY STATEMENT

All benchmark evaluation tools and datasets used in this work have been publicly released, and the data collection platform will also be open-sourced once fully refined. We additionally provide detailed documentation, experimental settings, and implementation instructions to facilitate reproducibility and ensure that other researchers can replicate and build upon our work.

## D    LLM Usage Statement

In this paper, the use of LLMs is limited to translation and language polishing. No assistance was sought from LLMs in developing ideas or designing specific methods.

## E    Prompts and Examples

Please act as a subject matter expert in {subject} to evaluate student responses based on the following requirements:

**[Evaluation Task]**
Conduct a professional assessment of the student's step-by-step solution according to the question information, reference answer, and scoring guidelines, and provide a structured evaluation result.

**[Scoring Guidelines]**
{score_guideline}
{few_shot_samples}

**[Evaluation Materials]**
Question content: {question}
Total score: {total}
Error types: {error_type}
Reference answer: {reference}
Analysis: {analysis}
Student's answer: {student_answer}

**[Evaluation Process and Requirements]**
**1. Step-by-step analysis**:
    - Break down each step of the student's solution.
    - Independently evaluate each step:
        * Determine correctness ('label').
        * If errors exist, select one or more primary causes from the error list ('errors').
    - Single-step evaluation format: {{'step_score': step score, 'errors': [error causes]}}
**2. Comprehensive assessment**:
    - Summarize the scores of each step to calculate the total score.
    - Provide an overall evaluation ('label').
**3. Result output**:
    - Use standard JSON format:
    {{
    'total': total score,
    'pred_score': evaluated total score,
    'steps': [step-by-step evaluation results]
    }}
    - 'pred_score' must be within the range of 'total'.
    - The cumulative value of 'step_score' must also be within the range of 0 to 'pred_score'.

Please complete the evaluation according to the above specifications and output the standardized assessment result in JSON format.

Figure 6: Prompts for model scoring and error cause prediction, where the descriptions are provided in Chinese.

In this section, we present the prompts used for model evaluation in Figure 6 and an annotated example of a math short-answer response in Figure 7. Our observations indicate that current LLMs exhibit limitations in numerical reasoning. In particular, when step-wise scores and error causes are positioned before the overall score in the output format, the model tends to produce inflated or biased overall scores. In contrast, generating the overall score (pred_score) first helps guide and constrain the subsequent step-level scoring and error analysis. Based on this insight, we design the output format to require models to first generate the overall score, followed by the evaluation and analysis of each individual step.

```
{
    "id": "Math_ShortAns_3",
    "question": "20. (10 points) Let $F_{1}, F_{2}$ be …",
    "reference": "Solution: (1) By the definition of an ellipse … (2) The equation of $L$ is $y= x+c$, where …",
    "analysis": "(1) By the definition of an ellipse,
$\\left|\\mathrm{AF}_{2}\\right|+|\\mathrm{AB}|+\\left|\\mathrm{BF}_{2}\\right|=4$. …",
    "total": 10,
    "steps": [
        {
            "response": "Let the foci of the ellipse be $F_{1}(-c,0)$ and $F_{2}(c,0)$, …",
            "label": "0",
            "errors": [
                "Calculation error"
            ]
        },
        {
            "response": "$$x^2 + \\frac{(k(x+c))^2}{b^2} = 1$$\nExpanding this equation …",
            "label": "0",
            "errors": [
                "Confused problem-solving approach",
                "Calculation error"
            ]
        },
        {
            "response": "$$x^2 + \\frac{(x+c)^2}{b^2} = 1$$",
            "label": "1",
            "errors": [
                "Correct step"
            ]
        },
        {
            "response": "Expanding:",
            "label": "0",
            "errors": [
                "Calculation error"
            ]
        },
        {
            "response": "$$x^2 + \\frac{x^2 + 2xc + c^2}{b^2} = 1…",
            "label": "0",
            "errors": [
                "Confused problem-solving approach",
                "Calculation error",
                "Incorrect or missing formula usage"
            ]
        },
        {
            "response": "",
            "label": "0",
            "errors": [
                "Confused problem-solving approach",
                "Calculation error"
            ]
        },
        {
            "response": "Final answers:\n(I) $|AB| = 2b$; (II) $b = 1$.",
            "label": 0,
            "errors": []
        }
    ],
    "manual_label": 1
}
```

Figure 7: Example format of a math short-answer response annotated with both the score and corresponding error causes.

## F   ANALYSIS OF ERROR CAUSE DETECTION CAPABILITIES

To assess whether models can accurately identify all relevant error causes in student responses, we evaluate their detection capabilities using the Micro-F1 score across different subjects and question types. For each question, we first merge and deduplicate the step-wise error causes predicted by the model and those annotated by human experts. We then compute true positives (TP), false positives (FP), and false negatives (FN) by checking the presence of each error cause in the predicted and reference sets. The Micro-F1 score is calculated based on the aggregated TP, FP, and FN across all questions. The results are summarized in Table 4.

Deepseek-R1 still achieves the highest average F1 score. However, compared to the ECS results in Table 3, all models demonstrate markedly higher F1 scores and narrower performance gaps. This suggests that while current LLMs are generally capable of identifying the overall set of error causes in a response, they still face challenges in accurately reasoning through step-wise, fine-grained errors

Table 4: Micro-F1 scores for error causes prediction (%) across various LLMs, evaluated by subject area and question type. **S.** indicates Short Answer, **M.** denotes Multiple Choice, and **G.** refers to Gap Filling. The best and second-best scores in each category are highlighted in **bold** and underlined, respectively.

| Models | Phy. (S.) | Phy. (M.) | His. (S.) | Geo. (S.) | Bio. (G.) | Chi. (G.) | Chi. (S.) | Math (S.) | Math (G.) | Pol. (S.) | Eng. (G.) | Che. (G.) | Avg. |
|---|---|---|---|---|---|---|---|---|---|---|---|---|---|
| | | | | | | Reasoning-based LLMs | | | | | | | |
| Deepseek-R1 | **51.82** | 45.71 | 44.47 | 43.12 | 65.60 | 83.33 | 62.35 | **65.19** | 54.67 | **60.47** | 62.46 | 56.07 | **57.94** |
| QwQ-32B | 45.81 | 47.73 | 44.30 | 41.38 | 59.19 | 66.07 | 58.82 | 64.84 | 44.63 | 56.09 | 58.89 | 53.91 | 53.47 |
| TinyR1-32B-Preview | 36.20 | 44.44 | 44.67 | 44.53 | 53.50 | 74.17 | 58.50 | 49.69 | 37.01 | 53.59 | 52.13 | 48.04 | 49.71 |
| Qwen3-32B | 46.72 | 47.13 | 41.66 | 40.29 | 59.06 | 81.18 | 57.34 | 59.53 | 48.98 | 54.57 | 57.61 | 52.21 | 53.86 |
| Qwen3-8B | 46.35 | **67.84** | 39.07 | 37.81 | 64.01 | 67.87 | 65.09 | 53.25 | 23.79 | 48.42 | 51.85 | 54.70 | 51.67 |
| MiMo-7B-RL | 43.42 | 48.45 | 31.49 | 36.79 | 53.09 | 86.21 | 44.81 | 35.45 | 49.59 | 20.51 | 46.11 | 27.96 | 43.66 |
| DeepSeek-Prover-V2-7B | 38.02 | 35.11 | 27.44 | 27.65 | 48.72 | 23.62 | 31.36 | 32.91 | 44.96 | 25.00 | 31.41 | 27.55 | 32.81 |
| DeepSeek-R1-Distill-7B | 34.90 | 47.95 | 36.36 | 27.57 | 47.36 | 42.70 | 29.55 | 26.98 | 51.85 | 38.85 | 50.32 | 19.65 | 37.84 |
| | | | | | | RLHF-based LLMs | | | | | | | |
| Deepseek-V3 | 48.58 | 46.93 | 41.79 | 42.52 | **67.79** | **92.24** | 62.36 | 63.05 | 53.59 | 52.30 | 65.60 | 56.68 | 57.79 |
| GPT 4o-mini-20240718 | 48.39 | 48.72 | **45.76** | 40.41 | 54.58 | 59.26 | 60.60 | 49.13 | 39.90 | 39.14 | 30.59 | 38.65 | 46.26 |
| Llama3.3-70B-Instruct | 49.57 | 45.88 | 42.55 | **45.54** | 58.79 | 67.10 | 53.92 | 51.72 | 43.27 | 47.70 | **69.64** | 30.68 | 50.53 |
| Mixtral 8×7B-Instruct | 33.67 | 43.39 | 41.58 | 37.86 | 52.41 | 33.72 | 58.86 | 37.57 | 59.28 | 41.66 | 41.53 | 48.58 | 43.34 |
| Qwen2.5-32B-Instruct | 45.00 | 45.71 | 40.75 | 41.75 | 62.60 | 89.63 | 54.27 | 61.39 | 42.67 | 53.85 | 48.37 | **56.77** | 53.56 |
| Qwen2.5-14B-Instruct | 43.70 | 45.24 | 39.86 | 36.82 | 61.73 | 88.26 | **65.34** | 48.14 | 49.49 | 49.17 | 54.75 | 50.43 | 52.74 |
| GLM4-9B-Chat | 40.00 | 44.32 | 41.10 | 37.14 | 55.79 | 48.31 | 57.48 | 38.76 | 38.73 | 42.35 | 41.50 | 42.42 | 43.99 |
| Llama3-8B-Instruct | 42.95 | 37.63 | 34.15 | 32.64 | 56.28 | 50.30 | 53.28 | 47.60 | **68.76** | 24.89 | 47.11 | 46.41 | 45.17 |

when compared to human annotations. The comparison between F1 and ECS further underscores the value of ECS as a metric. Unlike F1, which primarily reflects aggregate detection performance, ECS captures the consistency of reasoning across steps, offering a more nuanced and informative measure of model explainability in SAS tasks.

# G  ANALYZING THE RELATIONSHIP BETWEEN ECS AND SCORING PERFORMANCE

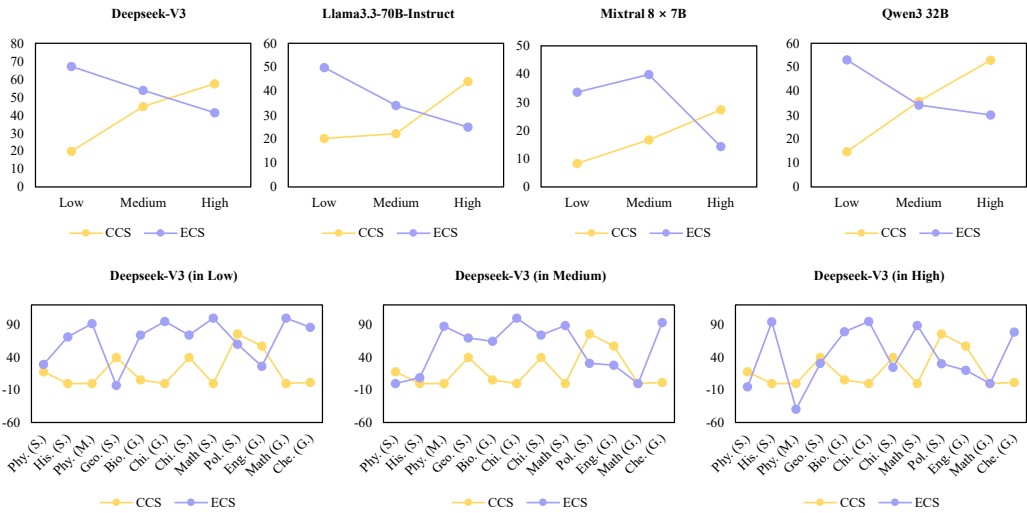

Figure 8: ECS and CCS comparison across low, medium, and high score intervals. The upper line chart illustrates the average ECS and CCS scores of different models across all subjects and question types, while the lower chart provides a detailed breakdown of Deepseek-V3's performance across individual subject areas and question types.

To further investigate the relationship between ECS and model scoring performance, we analyze the evaluation results of four representative models: Deepseek-V3, LLaMA3.3-70B-Instruct, Mixtral 8×7B, and Qwen3-32B. Following the score-based partitioning strategy described in §4.3, we divide the evaluation samples into three intervals: low, medium, and high based on the overall scores assigned by human annotators. For each interval, we compute the average CCS and ECS across all subjects and question types. The results are presented in Figure 8.

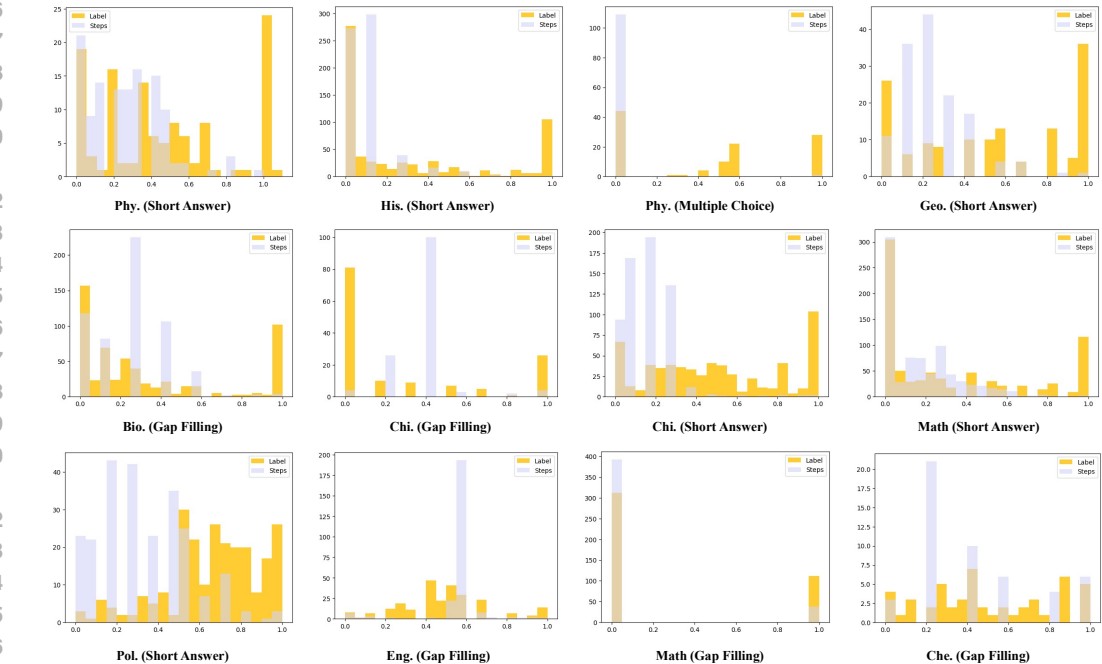

Figure 9: Normalized distributions of the ratio between total human-annotated scores and the number of steps in the dataset. For consistency, both the scores and step counts on the horizontal axis are presented in proportional form.

Notably, we observe an overall inverse trend between ECS and CCS: as ECS decreases, CCS tends to increase, and vice versa. To explore this phenomenon in more detail, we further examine Deepseek-V3's ECS and CCS across different intervals for each subject and question type. This finer-grained analysis confirms the inverse relationship, suggesting that a model's consistency in error cause identification may affect its overall scoring performance. Moreover, ECS values tend to drop more significantly in the high-score interval, indicating that models are more effective at identifying error causes in lower-scoring responses. One possible explanation is that models may over-focus on local error causes, leading to reduced leniency compared to human annotators and ultimately resulting in lower scoring consistency. When considered alongside the results in Table 4, which show that models achieve higher performance in overall error cause detection (as measured by Micro-F1) than in ECS, these findings suggest that the relatively low ECS scores are primarily attributable to the models' challenges in accurately reasoning about step-wise error causes in medium- and high-scoring responses.

## H  DATA DISTRIBUTION VISUALIZATION

To better understand the composition of our dataset, we visualize the distributions of human-annotated overall scores and the number of reasoning steps in student responses across different subject areas and question types, as shown in Figure 9. For clarity, both the scores and step counts on the horizontal axis are presented in proportional form.

As the figure illustrates, all subject-question types, except for **Math (G.)**, which consists of single-step gap-filling questions and thus only includes full or zero scores, span a broad range of scores across low, medium, and high intervals. While real-world student response distributions typically approximate a normal distribution, our dataset contains a larger proportion of responses with either full marks or zero scores. This is largely due to the data selection process. However, as described in § 3.2, we applied rigorous data cleaning to ensure high semantic diversity within each score range. This diversity is crucial for evaluating whether models can consistently and accurately score varied responses that fall within the same scoring band.

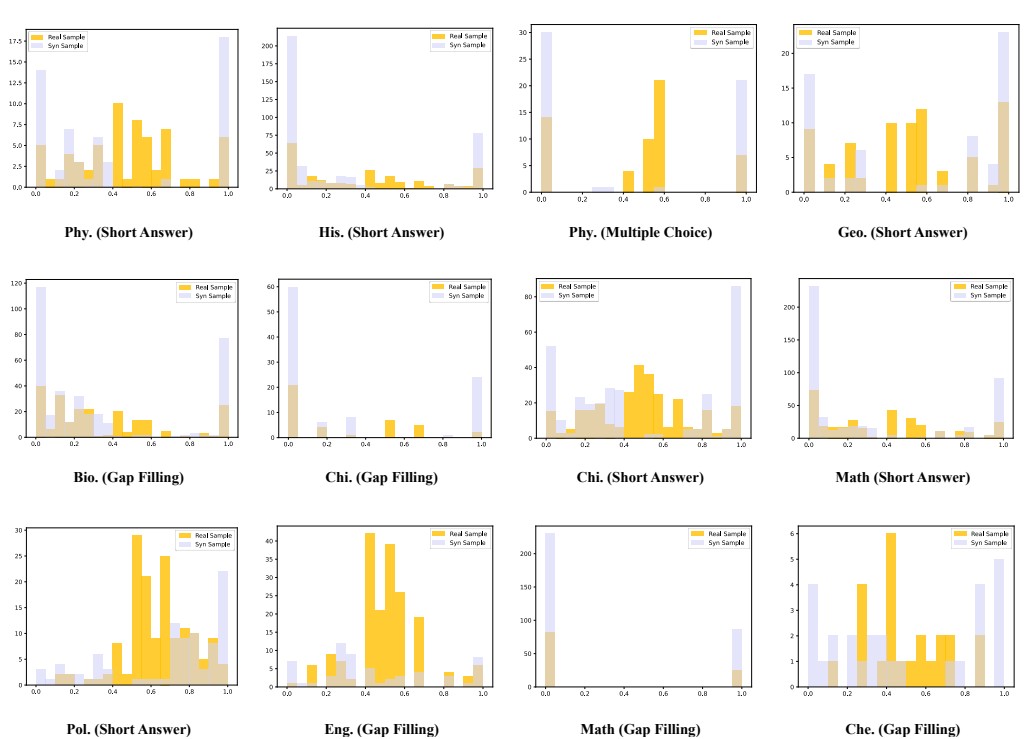

Figure 10: Distribution of human-annotated normalized scores for real student responses and LLM-synthesized responses.

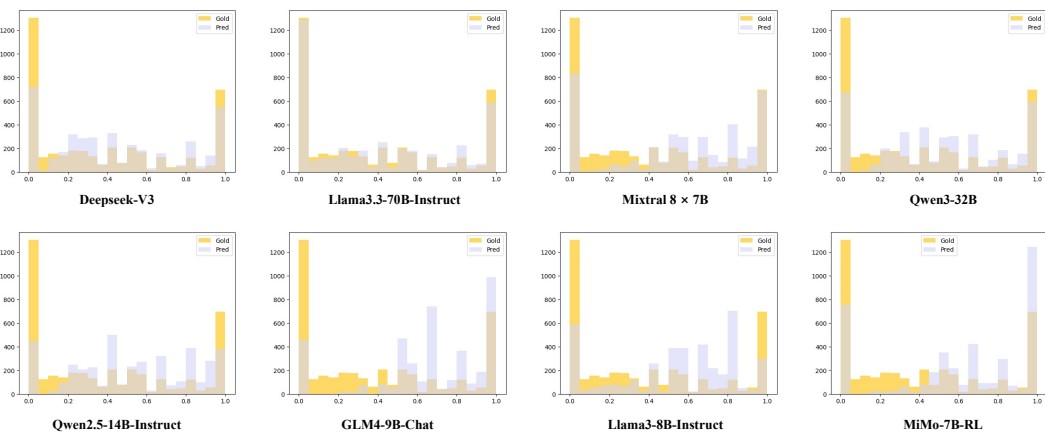

Figure 11: Normalized distributions of total human-annotated scores and the number of steps in the dataset.

To further assess the distributional validity of the synthesized data, Figure 10 compares the human-annotated score distributions of real student responses and LLM-generated responses. Real responses are largely concentrated in the mid-range interval of [0.4, 0.7], whereas synthesized responses appear more frequently at both the lower and higher ends. This pattern reflects two factors: real students tend to cluster around average performance levels, making it difficult to collect sufficiently diverse low- and full-score responses, while LLMs struggle to generate mid-range answers that capture the nuanced reasoning patterns of real students. By combining authentic responses with synthesized samples that introduce controlled errors or varied correct answers, we mitigate these limitations and obtain a more balanced overall score distribution.

## I  ANALYSIS OF PREDICTED SCORE DISTRIBUTIONS

We selected eight representative models of varying sizes and compared their predicted overall score distributions with human-annotated scores across the entire dataset, as illustrated in Figure 11.

Overall, model predictions tend to concentrate in the mid-score range, reflecting a more conservative scoring pattern compared to human annotators, models are generally less inclined to assign either full or zero scores. Notably, we also observe considerable variation across models in their scoring tendencies within this middle range. Among them, LLaMA3.3-70B-Instruct shows a score distribution most closely resembling that of human annotations. However, when we examine the step-wise evaluation metrics presented in Table 2 and Table 3, LLaMA3.3-70B-Instruct still lags behind Deepseek-V3 in both CCS and ECS performance. This discrepancy underscores the strength of our proposed evaluation metrics, which capture deeper aspects of model performance in SAS tasks, beyond mere alignment with overall scores.

## J  COMPARISON BETWEEN LLMs AND SLMs

In the introduction, we stated that LLMs underperform SLMs on short answer scoring when neither complex prompt design nor task-specific fine-tuning is applied. This claim was initially based on empirical observations without a bibliographic reference, and we now provide additional evidence to substantiate it. We conducted supplementary experiments on three representative SAS datasets: LE (Logistics Engineering short answer questions) (Lai et al., 2024), ASAG (Computer Programming questions) (Mohler & Mihalcea, 2009), and SR (Biomedical domain questions) (Menini et al., 2019). These datasets contain real student responses paired with human-assigned scores.

We designed two sets of experiments for SLMs. The first follows a full-shot setting, in which models are trained on all available data (LE 409, ASAG 440, SR 231). The second adopts a few-shot setting with an "N-way K-shot" configuration. Each question forms an N-way task, and K is set to 2, resulting in 200 samples for LE, 42 for ASAG, and 68 for SR. This setup allows for a comparison of model performance under different data regimes. We evaluated several representative SLMs, including vanilla BERT-base (Devlin et al., 2019), SPRAG (Bonthu et al., 2023), M-Sim (Lai et al., 2024), and ERNIE 3.0-base (Sun et al., 2021) with 0.11B parameters, and compared them with LLMs using zero-shot inference and full-shot training via supervised fine-tuning (SFT). The results are summarized in Table 5.

As shown in the table, several LLMs perform worse in the full-shot setting on ASAG and SR than in the zero-shot setting. A possible explanation is that increasing single-domain training data may exacerbate catastrophic forgetting on this relatively simple downstream scoring task, thereby reducing generalization ability. Overall, LLMs demonstrate advantages over SLMs in the few-shot setting, which is expected given the limited capacity of SLMs and their reliance on larger amounts of training data to achieve strong performance. However, even in the SLMs' few-shot setting, the strongest LLM in our experiments still underperforms ERNIE 3.0 on the LE and SR datasets. In the full-shot regime, the performance gap between LLMs and SLMs becomes more pronounced, with all SLMs substantially outperforming LLMs on ASAG and SR. These findings provide empirical support for our observation that LLMs do not consistently surpass SLMs on SAS tasks. While LLMs may be suitable for zero-shot scoring scenarios, their performance ceilings in supervised SAS remain an open challenge that warrants further investigation.

## K  ADDITIONAL BENCHMARK EVALUATION ON EXISTING DATASETS

In our main benchmark, we used LLM-generated responses to construct part of student responses, which may limit the benchmark's ability to fully reflect the challenges LLMs encounter in real grading scenarios, as synthetic responses can introduce distributional differences. To address this limitation, we additionally evaluate our benchmark framework on the LE, ASAG, and SR datasets, which contain real student responses with human-annotated scores. Following the step-wise construction and scoring methodology of our benchmark, we manually annotated and structured these datasets. We then evaluated several LLMs, and the results are presented in Table 6.

Table 5: Performance comparison of LLMs (SFT) and SLMs on SAS datasets in terms of QWK, with **bold** values indicating the best results. Standard deviations are provided in parentheses.

| Model | LE | ASAG | SR |
|---|---|---|---|
| Zero-shot | | | |
| Qwen3-8B | 61.74 (1.05) | 38.58 (1.44) | 42.69 (1.72) |
| Llama-8B-Instruct | 79.88 (1.81) | 37.91 (2.19) | 38.47 (1.58) |
| GLM4-9B-Chat | 82.23 (2.41) | 28.39 (2.77) | 51.07 (2.11) |
| Qwen2.5-32B-Instruct | 88.92 (2.73) | 69.41 (2.66) | 62.28 (2.12) |
| Qwen3-32B | 89.52 (1.48) | **70.76 (1.96)** | 69.85 (2.91) |
| Few-shot | | | |
| BERT | 88.98 (0.27) | 58.23 (0.14) | 33.18 (0.25) |
| SPRAG | 89.18 (0.39) | 58.18 (0.01) | 34.92 (0.21) |
| M-Sim | 89.24 (0.04) | 58.26 (0.20) | 41.87 (0.19) |
| ERNIE 3.0 | **93.24 (0.12)** | 58.08 (0.11) | **71.81 (0.23)** |
| Full-shot | | | |
| Qwen3-8B | 76.77 (2.14) | 37.81 (1.72) | 42.33 (2.89) |
| Llama-8B-Instruct | 84.22 (2.67) | 34.62 (1.35) | 47.03 (2.48) |
| GLM4-9B-Chat | 84.66 (1.91) | 30.01 (2.73) | 46.00 (1.64) |
| Qwen2.5-32B-Instruct | 91.80 (1.88) | 60.24 (2.31) | 64.49 (2.07) |
| Qwen3-32B | 91.88 (1.43) | 52.43 (2.56) | 55.61 (1.98) |
| BERT | 89.78 (0.05) | **75.35 (0.09)** | 67.48 (0.10) |
| SPRAG | 89.98 (0.04) | 75.28 (0.04) | 68.52 (0.01) |
| M-Sim | 90.18 (0.03) | 74.18 (0.04) | 68.34 (0.02) |
| ERNIE 3.0 | **92.61 (0.09)** | 73.49 (0.10) | **74.35 (0.07)** |

Table 6: Performance of different LLMs on the extended step-wise datasets. **Bold** indicates the best results. Standard deviations are provided in parentheses.

| Model | QWK | | | CCS | | | ECS | | |
|---|---|---|---|---|---|---|---|---|---|
| | LE | ASAG | SR | LE | ASAG | SR | LE | ASAG | SR |
| Qwen3-8B | 61.81 (3.12) | 39.45 (2.48) | 45.76 (3.67) | 60.11 (2.94) | 35.28 (3.41) | 40.07 (2.57) | 72.17 (7.82) | 46.52 (6.43) | 34.58 (9.14) |
| Llama3-8B-Instruct | 79.94 (2.77) | 34.96 (3.26) | 39.59 (2.61) | 78.82 (3.88) | 30.12 (3.11) | 32.01 (2.43) | 69.14 (5.72) | 43.32 (8.23) | 28.31 (6.94) |
| GLM4-9B-Chat | 82.11 (3.49) | 28.45 (2.83) | 51.99 (3.04) | 82.08 (3.22) | 25.53 (2.41) | 37.72 (3.15) | 70.28 (9.01) | 42.16 (7.36) | 32.98 (5.44) |
| Qwen2.5-32B-Instruct | 88.97 (2.66) | 69.34 (3.74) | 63.35 (3.18) | 87.16 (2.85) | **61.80 (3.07)** | **57.35 (2.68)** | 82.20 (6.41) | 65.53 (8.92) | **47.67 (7.33)** |
| Qwen3-32B | **89.46 (3.55)** | **70.83 (2.91)** | **69.92 (3.87)** | **89.09 (2.44)** | 60.73 (3.92) | 54.92 (3.28) | **88.75 (4.89)** | **67.97 (6.72)** | 42.29 (9.56) |

The results show that, for LLMs of comparable parameter sizes, performance varies across datasets. In most cases, CCS consistently remains lower than QWK, with overall trends aligning with our main experiments. These findings confirm that our proposed evaluation metrics effectively capture the inconsistent judgment exhibited by LLMs when scoring real student responses.

## L  DETAIL OF TEMPLATE-FREE RESPONSE DESIGN

Length and position biases in student responses are critical factors for evaluating the robustness of LLMs in scoring under natural variability. To address this, we explicitly incorporate such biases into the benchmark design. In constructing the dataset, although MCQs can typically be graded through rule-based methods, we represent them in a template-free format to better assess the genuine comprehension abilities of LLMs, as illustrated in Figure 12. Furthermore, gap-filling responses are converted into short-answer formats. Unlike directly filling in blanks, the short-answer format allows respondents to add complementary expressions more freely, thereby increasing the diversity of response forms. These transformations of MCQ and gap-filling responses enable us to evaluate how LLMs handle variations in response length and in the position of key information. The objective is to determine whether LLMs can maintain consistent scoring accuracy despite such surface-level differences. Any observed performance fluctuation in these cases can be attributed to bias sensitivity, which our benchmark is specifically designed to capture.

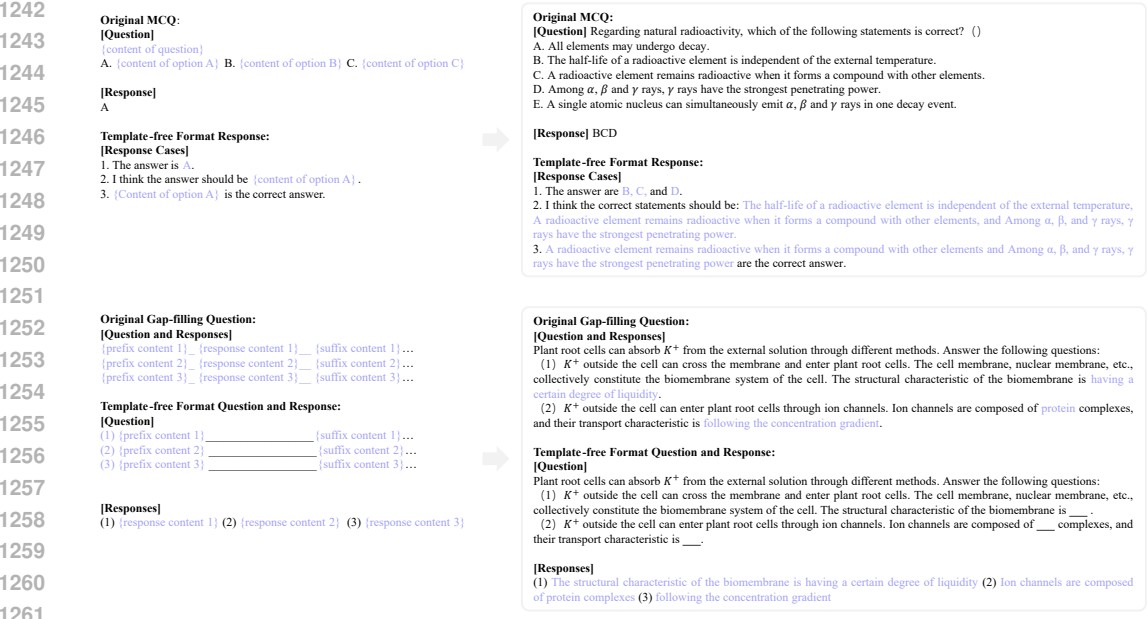

Figure 12: Examples of template-free conversions for MCQs and gap-filling questions. The figure on the right shows a specific question example, where variables or response content are highlighted in purple.

# M  ANALYSIS OF POSITION AND LENGTH BIASES

To further investigate how the benchmark reflects position and length biases, we conducted a series of error analysis experiments. For position bias, we examined samples from template-free MCQs, where the correct answer option was deliberately placed at different positions within the response. We then calculated the average accuracy of LLMs in identifying the correct option under these conditions, as presented in Table 7. It can be observed that most models achieve accuracy below 50% under the template-free format, indicating generally low performance. These results demonstrate that position bias has a substantial impact on scoring accuracy and suggest that many LLMs struggle to maintain robustness against positional variations in responses, which likely contributes to inconsistencies in performance across subjects.

Table 7: Average accuracy of different LLMs in position bias analysis. **Bold** values indicate the best results. Standard deviations are provided in parentheses.

| Model Type | Model | Avg. Accuracy (%) |
|---|---|---|
| **Reasoning-based LLMs** | Deepseek-R1 | **51.37 (1.92)** |
| | QwQ-32B | 37.83 (2.48) |
| | TinyR1-32B-Preview | 39.22 (1.63) |
| | Qwen3-32B | 41.57 (2.74) |
| | Qwen3-8B | 32.96 (1.57) |
| | MiMo-7B-RL | 23.92 (2.11) |
| | Deepseek-Prover-V2-7B | 21.08 (2.95) |
| | DeepSeek-R1-Distill-7B | 22.87 (3.04) |
| **RLHF-based LLMs** | Deepseek-V3 | **52.02 (2.33)** |
| | GPT-4o-mini (20240718) | 16.82 (1.67) |
| | Llama3.3-70B-Instruct | 46.19 (2.08) |
| | Mixtral 8×7B-Instruct | 24.91 (2.79) |
| | Qwen2.5-32B-Instruct | 36.27 (1.51) |
| | Qwen2.5-14B-Instruct | 38.46 (2.66) |
| | GLM4-9B-Chat | 23.83 (1.38) |
| | Llama3-8B-Instruct | 22.90 (2.57) |

To investigate length bias, we analyzed responses of varying lengths to the same question, selecting the longest, shortest, and medium-length responses, and compared the differences between the

average model-predicted scores and the corresponding human-annotated scores. Specifically, we examined both overall average scores and step-wise average scores across different response lengths, as shown in Figure 13. The results indicate that, for both gap-filling and MCQ tasks, the discrepancies between model predictions and human annotations change noticeably with response length, confirming that the template-free design effectively captures length bias. Furthermore, deviations in step-wise scores are generally more pronounced than those in overall scores, regardless of question type. In particular, as response length increases, step-wise predictions diverge more substantially from human annotations than overall score predictions. These findings highlight the necessity of our step-wise design for effectively detecting length bias.

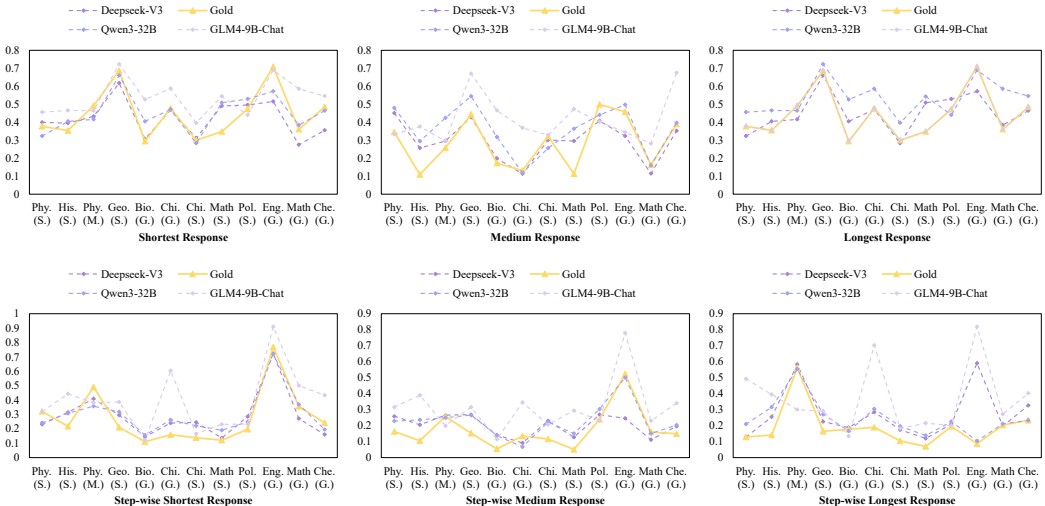

Figure 13: The differences between LLM-predicted scores and human-annotated scores for responses of varying lengths. "Gold" denotes the human-annotated scores. The top three panels display the differences in overall average scores, while the bottom three panels depict the differences in step-wise average scores.

## N  COMPLEXITY OF THE DATASET

To analyze the complexity of responses, we computed the average semantic similarity between answers within the same score band and across different questions using the "text2vec-base-multilingual" (Xu, 2023) model. We assume that lower similarity indicates higher diversity and, consequently, greater complexity in the responses. We also report the maximum number of steps per subject. The results are summarized in Table 8. By comparing the experimental results in Table 2 and Table 3, we observe that subjects with a greater number of steps and lower semantic similarity tend to be more challenging for LLMs, as they demand more complex reasoning and exhibit higher diversity in student responses.

## O  STEP DIVISION

To enable fine-grained evaluation, we develop step division rules tailored to different question types. Representative examples of each case are provided in Figure 14. Overall, the division of response steps can be categorized into three types:

Case 1 (Reasoning and Calculation Based). This category includes subjects such as Mathematics (S.) and Physics (S.). These questions require systematic reasoning and step-by-step calculations. Responses are therefore segmented into steps according to key reasoning nodes.

Case 2 (Argument Based). This category includes subjects such as Geography (S.), parts of Politics (S.), and Chinese (S.). Such questions typically call for discursive answers, which are divided into steps based on distinct arguments or supporting evidence.

Table 8: Maximum number of steps and semantic similarity across subjects. "Sim." represents the semantic similarity scores.

| Subject | Max Steps | Sim. (↑ easier, ↓ harder) |
|---|---|---|
| Physics (S.) | 16 | 66.02 |
| Physics (M.) | 2 | 63.62 |
| History (S.) | 8 | 52.39 |
| Geography (S.) | 10 | 53.50 |
| Biology (G.) | 8 | 62.73 |
| Chinese (G.) | 6 | 70.65 |
| Chinese (S.) | 12 | 50.87 |
| Math (S.) | 25 | 64.73 |
| Math (G.) | 2 | 59.73 |
| Politics (S.) | 12 | 60.59 |
| English (G.) | 16 | 77.58 |
| Chemistry (G.) | 8 | 67.35 |

# Case 1

[Question]

Let an arithmetic sequence $\{a_n\}$ satisfy $a_3 = 5$ and $a_{10} = -9$

(I) Find the formula for the general term of $\{a_n\}$

(II) Find the sum $S_n$ of the first $n$ terms of $\{a_n\}$ and the value of the index $n$ for which $S_n$ is maximized.

[Response]

STEP 1: From $a_n = a_1 + (n-1)d$ and given $a_3 = 5, a_{10} = -9$, we have:
$$\begin{cases} a_1 + 2d = 5 \\ a_1 + 9d = -9 \end{cases}$$

STEP 2: Solving these equations, we get $d = -2, a_1 = 9$.

STEP 3: The formula for the general term of the sequence $\{a_n\}$ is $a_n = 11 - 2n$.

STEP 4: We know that $S_n = na_1 + \frac{n(n-1)}{2}d = 10n - n^2$.

STEP 5: Since $S_n = -(n-5)^2 + 25$,

STEP 6: therefore, when $n = 5$, $S_n$ reaches its maximum value.

# Case 2

[Question]

Driven by economic interests, people clear tropical rainforests to plant tropical cash crops such as rubber and oil palm. Observations have shown that in some areas, the humidity within artificially planted rubber plantations is significantly lower than in rainforests. Analyze the reasons for the lower humidity in these artificially planted rubber plantations compared to rainforests.

[Response]

STEP 1: Rubber trees have a higher water demand than rainforests, exhibit strong transpiration, and cause significant water loss.

STEP 2: With relatively low plant density, air circulates more rapidly through rubber plantations, accelerating moisture dissipation and leading to land drying; this reduces the amount of water vapor evaporated from the soil into the air.

STEP 3: Rubber trees have a weaker capacity for soil and water conservation and water retention, thereby affecting the ecological environment and climate change.

STEP 4: The artificially planted rubber plantation ecosystem is a monoculture with lower biomass per unit area, which is unfavorable for maintaining air humidity.

# Case 3

[Question]

The tundra ecosystem is unique due to its harsh living conditions for organisms, and the region where this ecosystem is located has been called a "barren land." Answer the following questions: (1) Due to the limiting effects of temperature, species richness on the tundra is relatively low. Richness refers to ___. (2) Compared to tropical forest ecosystems, the tundra ecosystem generally favors the accumulation of soil organic matter. The reason for this is ___. (3) Typically, food chains in ecosystems are not very long. The reason for this is ___.

[Response]

STEP 1: Richness refers to the number of species in a community.

STEP 2: The reason is that compared with tropical forest ecosystems, the temperature in tundra ecosystems is lower, which is not conducive to the decomposition of soil organic matter by microorganisms in the soil and is conducive to the accumulation of soil organic matter.

STEP 3: The reason is that energy decreases step by step as it flows along the food chain.

Figure 14: Examples of different step division cases.

Case 3 (Pre-set Question Structure). This category includes subjects such as Mathematics (G.), Politics (S.), Chinese (G.), Chinese (S.), English (G.), Physics (G.), Biology (G.), History (S.), and Chemistry (G.). In these cases, the step structure is already defined by the design of the question, and responses are segmented directly according to the sequence of sub-questions.

# P   HUMAN EVALUATION

Table 9: Average human performance across all evaluation metrics on SAS-Bench.

| Metric | Phy. (S.) | Phy. (M.) | His. (S.) | Geo. (S.) | Bio. (G.) | Chi. (G.) | Chi. (S.) | Math (S.) | Math (G.) | Pol. (S.) | Eng. (G.) | Che. (G.) | Avg. |
|---|---|---|---|---|---|---|---|---|---|---|---|---|---|
| QWK | 87.72 | 96.59 | 88.63 | 98.87 | 97.45 | 95.64 | 94.73 | 95.84 | 93.81 | 90.64 | 92.19 | 92.86 | 93.75 |
| CCS | 85.74 | 95.39 | 88.02 | 97.08 | 96.57 | 92.39 | 92.06 | 94.75 | 92.37 | 90.38 | 86.81 | 95.52 | 92.26 |
| ECS | 86.77 | 89.45 | 93.81 | 93.79 | 92.83 | 93.14 | 94.30 | 98.10 | 90.73 | 90.67 | 87.29 | 96.40 | 92.27 |

To provide an objective reference for evaluating LLM performance, we report human results in Table 9. For each subject, two human experts independently scored the responses according to the scoring guidelines, and the final score was computed as the average of their annotations. Human performance exceeds 90% on QWK, CCS, and ECS. Although consistency on short-answer questions is slightly lower than that of other question types, it still remains above 85% on CCS. These results highlight a notable gap between LLMs and human evaluators in scoring consistency.

**Case 1**

**[Question] (Standard Short Answer Question):**

Guxiang Gully is located in the southeastern part of the Qinghai–Tibet Plateau. There are six glaciers in the upper reaches, and the valley contains more than 300 meters of thick deposits. The middle section of the river valley is narrow, and the glacier tongues extend deep into the forest belt. Glacier-related debris flows frequently erupt in summer and autumn, and their destructive power is rarely seen anywhere else in the world. Analyze the reasons why glacier debris flows in Guxiang Gully exhibit extremely high destructive power during summer and autumn.

**Full Score:** 10

**Reference Answer:**

The thick loose deposits provide an exceptionally abundant source of material; high temperatures in summer and autumn lead to intense glacier melting, and precipitation is also plentiful during this period. The combination of meltwater and rainfall offers an extremely abundant water supply; the narrow valley and large elevation drop allow energy to accumulate to a great extent.

**[Response Case A]:**

**STEP 1:** The tremendous destructive power of glacier debris flows in Guxiang Gully during summer and autumn results from the combined effects of an exceptionally abundant supply of loose materials, an extremely ample water source, and topographical conditions that facilitate energy accumulation. The thick loose deposits in the valley, reaching over 300 meters, provide a vast reservoir of solid material and form the fundamental premise for debris flow hazards. Concentrated heavy rainfall in summer and autumn, together with intense glacier melting caused by high temperatures, produces an abundant water supply that not only lubricates and destabilizes the deposits but also saturates and mobilizes large quantities of loose material. The narrow middle valley strongly constrains the debris flow, causing continuous kinetic energy accumulation and a sharp increase in flow velocity. Together with the large terrain drop, this further transforms into powerful impact and erosive forces. Therefore, the coupling of abundant material sources, overlapping water sources, and energy-accumulating topography jointly leads to the exceptionally intense glacier debris flows in Guxiang Gully during summer and autumn.

**Manual Score:** 10   **Deepseek-V3:** 10   **Qwen2.5-14B-Instruct:** 10

**[Response Case B]:**

**STEP 1:** There are six glaciers in the upper reaches, and during summer and autumn, both glacier meltwater and precipitation are substantial, providing sufficient water conditions. The valley contains over 300 meters of deposits, offering abundant material. The fact that "Guxiang Gully is located in the southeastern Qinghai–Tibet Plateau" suggests steep terrain. The narrow valley, large elevation drop, and rapid water flow enable significant energy accumulation for debris-flow formation. In summary, the glacier debris flows in Guxiang Gully have great destructive power during summer and autumn.

**Manual Score:** 10   **Deepseek-V3:** 9   **Qwen2.5-14B-Instruct:** 7

**[Response Case C]:**

**STEP 1:** Summer and autumn bring abundant meltwater and rainfall; the valley contains rich deposits; and the terrain is steep, with a narrow valley and large elevation drop that together allow debris flows to accumulate energy.

**Manual Score:** 9   **Deepseek-V3:** 6   **Qwen2.5-14B-Instruct:** 5

**[Response Case D]:**

**STEP 1:** The steep terrain, narrow valley, and large elevation drop allow debris flows to accumulate energy, and the valley contains abundant deposits, with significant meltwater and rainfall during summer and autumn.

**Manual Score:** 9   **Deepseek-V3:** 5   **Qwen2.5-14B-Instruct:** 4

Figure 15: Comparison between LLM and human scoring on standard short-answer questions across different response lengths.

# Q   BIAS ANALYSIS CASE STUDY

Figures 15 and 16 present two sets of student responses used to examine model biases. Case 1 contains responses of varying lengths that are correct or nearly correct, while Case 2 includes responses that are almost entirely incorrect, with differences in both length and the ordering of answer components. We compare the scoring behaviors of DeepSeek-V3 and Qwen2.5-14B-Instruct across these cases.

In Case 1, both models tend to assign higher scores to longer responses, even when shorter ones are also nearly correct according to human evaluators. We further find that inconsistent placement of

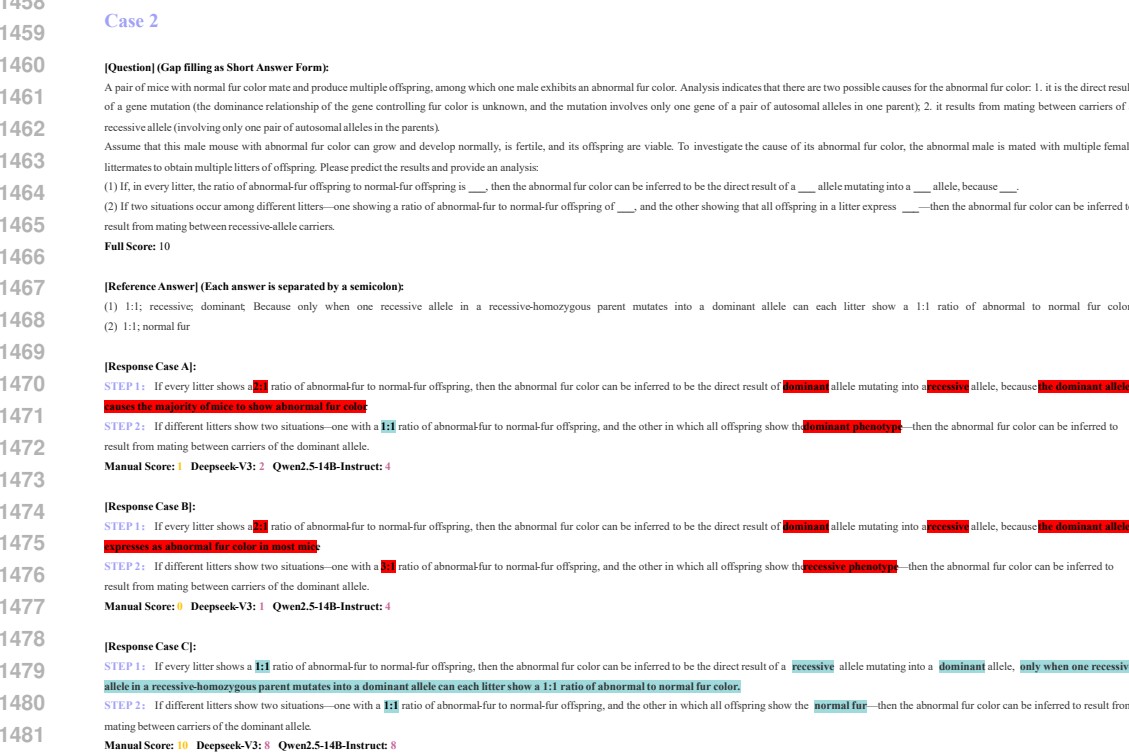

Figure 16: Comparison between LLM and human scoring on template-free gap-filling questions when incorrect segments (highlighted in red) are injected at different positions within extended responses, with correct segments highlighted in green.

key information, as seen in the responses of students C and D, can result in substantial variation in the scores produced by the models. Case 2 focuses on template-free gap-filling questions in which fully incorrect answers are embedded within lengthy responses. In Case B, we swapped the positions of the terms "dominant" and "recessive," and observe that Qwen2.5-14B-Instruct frequently overestimates such responses and assigns inflated scores. DeepSeek-V3 performs more robustly in this scenario, which is consistent with the patterns reported in the main experimental results in Table 2. These results indicate that current LLMs remain sensitive to response length and the placement of key information. In combination with the findings in Table 2, the results suggest that template-free gap-filling mitigates these biases more effectively for advanced LLMs on average. However, small-parameter LLMs are still substantially affected by these sources of bias.

Table 10: Pearson correlation coefficients between CCS and QWK across different subjects and question types, along with a comparison of QWK scores with and without step-wise scoring ($\Delta$).

| Model | Pearson R | $\Delta$ |
|---|---|---|
| Deepseek-R1 | 0.84 | -1.31 |
| QwQ-32B | 0.62 | 2.78 |
| TinyR1-32B-Preview | 0.899 | 2.45 |
| Qwen3-32B | 0.485 | 2.91 |
| Qwen3-8B | 0.222 | -2.67 |
| MiMo-7B-RL | 0.56 | 2.12 |
| Deepseek-Prover-V2-7B | 0.879 | -2.98 |
| DeepSeek-R1-Distill-7B | -0.124 | 3.27 |
| Deepseek-V3 | 0.633 | 1.85 |
| GPT 4o-mini-20240718 | 0.272 | -1.23 |
| Llama3.3-70B-Instruct | 0.93 | 2.46 |
| Mixtral 8×7B-Instruct | 0.74 | -2.89 |
| Qwen2.5-32B-Instruct | 0.493 | 2.55 |
| Qwen2.5-14B-Instruct | 0.688 | -2.77 |
| GLM4-9B-Chat | 0.451 | 2.33 |
| Llama3-8B-Instruct | 0.313 | -3.99 |

## R    IMPACT ANALYSIS OF INTRODUCING STEP-WISE SCORING

To further quantify the relationship between CCS and QWK, as well as the effect of incorporating step-wise scoring on overall performance, as shown in 10, we computed the Pearson correlation between CCS and QWK across subjects for different models. Most models show a positive correlation, and more than half demonstrate a strong positive correlation. As discussed in Experiment 4.2, a weak positive correlation suggests that a model may achieve high consistency on overall scores while exhibiting lower consistency at the step level. This does not indicate instability in QWK.

We also compared QWK scores with and without step-wise scoring. The difference, reported as the $\Delta$ value, falls within the normal range of variation and includes minor increases and decreases. These results indicate that the introduction of step-wise evaluation has only a minimal impact on overall scoring performance.

