# OpenReview forum: "SAS-Bench: A Fine-Grained Benchmark for Evaluating Short Answer Scoring with Large Language Models"
_ICLR.cc/2026/Conference — Submitted to ICLR 2026_

### Official Review · Reviewer_5Cgo · 2025-10-17

**Soundness:** 3
**Presentation:** 2
**Contribution:** 3
**Rating:** 6
**Confidence:** 4

**Summary:**

The paper introduces a new dataset and evaluation framework for automated short-answer scoring (SAS). The dataset comprises short-answer questions and student answers from China’s national university entrance exam, each annotated by experts with scores and error reasons. Extensive experiments show that current LLMs struggle with consistent step-wise scoring, especially on science questions, and that providing scoring rubrics and examples improves performance.

**Strengths:**

- The paper addresses a gap in SAS by introducing two new evaluation metrics: step-wise scoring and error-cause annotations. While prior benchmarks only provide an only score, these new metrics allow for an evaluation of a model’s reasoning process and explanation quality.
- The SAS-Bench dataset is substantial, well-annotated, and anticipated to be open-source. It contains diverse question types (science, English, math proofs, etc.) and expert annotations.
- The experiments include diverse comparisons (e.g., analyses of model size, reasoning approach, and training paradigm) yields important insights. For example, one insight from comparing different model sizes is that without complex prompt engineering or task-specific fine-tuning, LLMs tend to underperform smaller LMs in SAS.

**Weaknesses:**

- More differentiation is needed from existing SAS benchmark datasets such as the Kaggle ASAP-SAS and SemEval-2013 “Student Response Analysis” corpora datasets. The former dataset is introduced in the Related Work but not compared explicitly with SAS-Bench, and the paper associated with the latter dataset (Myroslava et al., 2013) is in the reference list but not discussed at all in the paper. The authors claim SAS-Bench is the first benchmark specifically tailored for SAS with LLMs, but they do not articulate how its novelty goes beyond combining known elements (e.g., reference answers and multidimensional scoring) in a new dataset.
- The performance tables and charts should include a measure of variance. Since LLM outputs can be stochastic, reporting single-run results without any measure of uncertainty means we can’t tell if differences between models are significant or just noise. Similarly, the authors do not report inter-annotator agreement in the form of Cohen's kappa coefficient for their expert labels.
- The SAS-Bench dataset, while diverse in subjects, is narrow in source. This raises concerns about how the benchmark’s insights would generalize to other contexts; e.g., different countries’ exams, free-form vs. structured answers, etc. The paper does not discuss whether models performing well on SAS-Bench would generalize to other curricula or languages.
- A portion of student responses were synthetically granted by LLMs, and then annotated, to augment real answers. The paper should acknowledge that relying on LLM-synthesized data could introduce biases, for example, if LLMs recognize their own style or content in these answers.

**Questions:**

- Most questions in the dataset are in Chinese (except the English category). Are some of the models evaluated better suited for Chinese language questions?
- What differentiates, if anything, a “short answer” from a close-ended exact-match answer, such as those in the HLE dataset (arXiv:2501.14249v9)?
- How does the LLM-as-a-Judge framework used in the paper compare to previous works?

---

> ### Author Response · Authors · 2025-11-21
>
> **Response W1:** Thank you for the comment. Prior work on SAS and related Automated Essay Scoring tasks has consistently shown that LLM performance in zero-shot and few-shot settings remains inadequate, as reported in [1], [2], and [3]. However, these studies primarily focus on evaluating overall scores and rarely examine model behavior through bias analysis or other diagnostic perspectives. This limitation arises because existing datasets typically include only questions, reference answers, human scores, and categorical labels, which restricts the ability to conduct detailed investigations of scoring behavior.
>
> A key motivation for our benchmark is therefore to introduce step-wise scores and error-cause annotations. These additions enable more fine-grained evaluation of LLM scoring capabilities and allow systematic analysis of potential biases. By providing this richer annotation framework, SAS-Bench goes beyond simply combining elements from previous datasets and offers a more comprehensive tool for understanding and improving LLM-based scoring. This is why we situate our motivation within the broader literature on LLMs as automated judges.
>
> [1] LLMs in Short Answer Scoring: Limitations and Promise of Zero-Shot and Few-Shot Approaches, 2024
>
> [2] Is GPT-4 Alone Sufficient for Automated Essay Scoring, 2024
>
> [3] Unleashing Large Language Models’ Proficiency in Zero-Shot Essay Scoring, 2024
>
> **Response W2:** Thank you for the comment. We have updated Tables 2 and 3 in the main experiments to include standard deviations in parentheses, which reflect the variability in model performance. In addition, we provide the average results from two human experts to illustrate inter-annotator consistency.
>
> |            |               |               |               |               |               |               |               |               |               |               |               |               |          |
> | ---------- | ------------- | ------------- | ------------- | ------------- | ------------- | ------------- | ------------- | ------------- | ------------- | ------------- | ------------- | ------------- | -------- |
> | **Metric** | **Phy. (S.)** | **Phy. (M.)** | **His. (S.)** | **Geo. (S.)** | **Bio. (G.)** | **Chi. (G.)** | **Chi. (S.)** | **Math (S.)** | **Math (G.)** | **Pol. (S.)** | **Eng. (G.)** | **Che. (G.)** | **Avg.** |
> | QWK        | 87.72         | 96.59         | 88.63         | 98.87         | 97.45         | 95.64         | 94.73         | 95.84         | 93.81         | 90.64         | 92.19         | 92.86         | 93.75    |
> | CCS        | 85.74         | 95.39         | 88.02         | 97.08         | 96.57         | 92.39         | 92.06         | 94.75         | 92.37         | 90.38         | 86.81         | 95.52         | 92.26    |
> | ECS        | 86.77         | 89.45         | 93.81         | 93.79         | 92.83         | 93.14         | 94.30         | 98.10         | 90.73         | 90.67         | 87.29         | 96.40         | 92.27    |
>
> **Response W3 and Q1:** Thank you for the suggestion. The primary goal of SAS-Bench is to evaluate whether LLMs can align with human scoring when provided with the question, reference answer, and scoring guideline. In addition, the benchmark enables fine-grained assessment of step-wise score consistency and error-cause consistency, offering insights into the interpretability of model scoring. Consequently, the dataset focuses on commonly taught middle-school subjects.
>
> Regarding language generalization, we have created an English version of the dataset using GPT-4o for translation, followed by human verification. The English version will be released once verification is complete.
>
> For free-form open-ended questions, the absence of clearly defined reference answers presents additional challenges in dataset construction compared to closed-ended questions; this will be addressed in future work. Regarding language bias, the current dataset naturally allows more accurate evaluation of model performance in Chinese. While the English version may not measure performance as precisely, it still enables a relative comparison of model capabilities.

---

> ### Author Response · Authors · 2025-11-21
>
> **Response W4:** Thank you for the comment. We include LLM-generated responses because real student answers tend to cluster in the medium to high score range. This portion of the data is also the most challenging to synthesize due to its diverse and authentic error patterns. LLMs, however, can reliably rewrite reference answers to produce high-scoring responses and generate low-scoring responses with clear, intentional errors. These synthetic samples help balance the dataset, compensating for the generally higher performance of real students. To ensure coverage across the full spectrum of ability levels, we use LLMs to generate both high- and low-scoring responses. Appendix H, Figure 11 presents the score distributions of synthetic and real responses, showing that real responses primarily fall between approximately 0.4 and 0.7, while LLM-generated responses effectively populate the high and low ends of the distribution.
>
> Regarding potential bias, both human and model scores are based on the same set of responses and follow the established scoring guidelines, so the results remain comparable. If a model assigns a higher or lower score than the human reference for a given sample, this indicates a misalignment in the model’s scoring ability. Capturing such discrepancies is a key objective of our benchmark.
>
> **Response Q2:** Thank you for the question. We note that the HLE dataset primarily evaluates a model’s ability to generate answers to questions, whereas the SAS dataset focuses on assessing a model’s capability as a grader, evaluating and scoring student responses according to established guidelines.
>
> **Response Q3:** Thank you for the question. Previous SAS datasets do not include step-wise scores or error-cause annotations. Besides, there is currently no unified or efficient LLM-as-a-Judge baseline for the SAS task. In our work, we design prompts specifically for step-wise evaluation and adopt an inference-only setup to ensure a fair and consistent assessment of each model’s performance.

---

### Official Review · Reviewer_aiMU · 2025-10-17

**Soundness:** 4
**Presentation:** 4
**Contribution:** 2
**Rating:** 4
**Confidence:** 4

**Summary:**

This paper introduces a new benchmark for Short Answer Scoring (SAS), called SAS-Bench. In SAS, short answer to questions seen in standardized tests are compared to a reference answer. Previous SAS benchmarks lack 2 important attributes, which this paper addresses: firstly, the assesment is not fine grained: in SAS-Bench, answers are broken down to steps using expert annotators. Secondly, each question has a predefined set of possible errors, which the expert annotators use to label the answer steps. Put together, this makes for a benchmark that allows for fine grained analysis of LLM as a judge performance for the SAS task.

**Strengths:**

- The paper is clearly written and well-structured.
- The figures are clean and easily understandable.
- The evaluation is thorough; a large quantity of reasoning and non-reasoning methods are tested along varied metrics that capture both LLM and expert consistency (CCS), as well as LLM error consistency (ECS).
- The benchmark and dataset is made up of multiple domains, with many examples: over 1,000 questions as well as over 4,000 expert annotations.

**Weaknesses:**

- The motivation is unclear to me: in the introduction, works like Zuang et al., 2024, Deshpande et al., 2024, and Raina et al., 2024 are cited, as works that show the shortcomings of LLMs-as-judges. As far as I can tell, none of these works deal with SAS, which results in the question: why are these works used as motivation for creating an SAS benchmark? Additionally, the gap presented in Appendix J also seems somewhat insignificant.  In general, it seems like LLMs already perform relatively well (see next point) on the benchmark, which questions the importance of SAS-Bench to the community.
- Looking at the consistency between experts and LLMs (CCS), it seems like the top model, V3, has a score of over 74\%. This suggests that the benchmark is already nearly saturated, which puts into question the strength of the contribution.
- There seems to be some inherent ambiguity in the construction of the expert annotations. For example, line 264 mentions that experts had disagreements that were resolved in discussions to reach a consensus. It seems like the way to segment an answer into steps, score and label each step could be done in several different reasonable ways, which varies from person to person.

**Questions:**

1. Are there examples of biases/shortcomings that LLMs exhibit on SAS? Is it possible to quantify them?
2. Is SAS-Bench saturated, or is there still progress left to be made by LLMs on it? Can you give concrete examples of interesting cases where LLMs are wildly inconsistent with humans on SAS-Bench?
3. Could there be differences of opinions regarding the way answers are segmented/scored/labeled from expert to expert? Other than discussions, is there a more well-defined way to resolve those differences?

---

> ### Author Response · Authors · 2025-11-21
>
> **Response W1:** Thank you for the comment. The studies by Zuang et al. (2024), Deshpande et al. (2024), and Raina et al. (2024) are cited in the introduction to illustrate broadly recognized limitations of LLM-as-a-judge methods in general evaluation settings. Although these works do not focus specifically on short-answer scoring, SAS is a finer-grained scoring task that inherits the same fundamental issues identified in the broader judge literature. Therefore, these studies help establish the general motivation for examining the reliability of LLM-based scoring systems.
>
> Moreover, prior work on SAS and related Automated Essay Scoring tasks has consistently shown that LLM performance in zero-shot and few-shot settings remains inadequate, as reported in [1], [2], and [3]. However, these studies primarily assess overall scores and seldom investigate model behavior through bias analysis or other diagnostic perspectives. One major reason is that existing datasets typically include only questions, reference answers, human scores, and categorical labels, which limits the ability to conduct detailed examinations of scoring behaviour. For this reason, a key motivation of our benchmark is to introduce step-wise scores and error-cause annotations that enable more fine-grained evaluation of LLM scoring capabilities and more systematic analyses of potential biases. This is also why we ground our motivation in the broader LLM-as-a-judge literature.
>
> The results in Appendix J further show that even with full-data supervised fine-tuning, a 32B model does not necessarily surpass a 0.11B baseline such as BERT-base or ERNIE3.0-base. In practice, fine-tuning very large models with at least 32B parameters is expensive, and these models are often reserved for inference rather than task-specific training. At the same time, smaller supervised models below 32B do not show clear performance gains over strong small language models and may suffer from reduced generalization due to catastrophic forgetting. These findings emphasize the need for a benchmark that can more accurately reveal the limitations and biases of LLM-based scoring systems.
>
> [1] LLMs in Short Answer Scoring: Limitations and Promise of Zero-Shot and Few-Shot Approaches, 2024
>
> [2] Is GPT-4 Alone Sufficient for Automated Essay Scoring, 2024
>
> [3] Unleashing Large Language Models’ Proficiency in Zero-Shot Essay Scoring, 2024
>
> **Response W2:** Thank you for the comment. Although high consistency between LLMs and human raters is indeed the ultimate goal of short-answer scoring, current models are still far from meeting the standards required in real-world educational settings. For instance, in the Chinese college entrance examination, the score difference between two human raters must remain within two points on a ten-point question. Any larger discrepancy triggers adjudication by a third expert. This practice reflects the extremely high level of agreement needed for dependable automated scoring. To contextualize this requirement, we report the average agreement between two human raters, where both QWK and CCS exceed 90%. Even for the best-performing model, DeepSeek V3, the average CCS is seventy-four, which still indicates a considerable gap. Current LLMs therefore cannot yet serve as reliable substitutes for human evaluators.
>
> Moreover, Table 2 shows that the tasks where models perform relatively well are gap filling and multiple choice questions, both constructed in a template-free manner. These subsets are primarily used to examine the influence of response-style bias. Larger models are less sensitive to such bias and can therefore align more closely with the reference solutions. In contrast, for short-answer questions, the average CCS drops to 67.50, which is much lower than the overall performance. This demonstrates that short-answer scoring remains a challenging scenario for current models.
>
> Our benchmark evaluates overall scoring accuracy while enabling fine-grained, interpretable analysis to identify error causes, providing insights to support more robust and trustworthy scoring systems.
>
> | | | | | | | | | | | | | | |
> | ---------- | ------------- | ------------- | ------------- | ------------- | ------------- | ------------- | ------------- | ------------- | ------------- | ------------- | ------------- | ------------- | -------- |
> | **Metric** | **Phy. (S.)** | **Phy. (M.)** | **His. (S.)** | **Geo. (S.)** | **Bio. (G.)** | **Chi. (G.)** | **Chi. (S.)** | **Math (S.)** | **Math (G.)** | **Pol. (S.)** | **Eng. (G.)** | **Che. (G.)** | **Avg.** |
> | QWK | 87.72 | 96.59 | 88.63 | 98.87 | 97.45 | 95.64 | 94.73 | 95.84 | 93.81 | 90.64 | 92.19 | 92.86 | 93.75 |
> | CCS | 85.74 | 95.39 | 88.02 | 97.08 | 96.57 | 92.39 | 92.06 | 94.75 | 92.37 | 90.38 | 86.81 | 95.52 | 92.26 |
> | ECS | 86.77 | 89.45 | 93.81 | 93.79 | 92.83 | 93.14 | 94.30 | 98.10 | 90.73 | 90.67 | 87.29 | 96.40 | 92.27 |

---

> ### Author Response · Authors · 2025-11-21
>
> **Response W3 and Q3:** Thank you for the comment. In science-related subjects, the scoring criteria are typically grounded in well-established teaching standards and scientific facts. Elements such as the correctness of formulas or the soundness of underlying principles are objective. Ambiguities, when they arise, usually concern the interpretation of specific criteria. In these situations, experts hold discussions and refine the scoring guideline until a shared understanding is reached.
>
> For humanities subjects, credit assignment also follows subject-specific teaching guidelines together with the reference answers. When disagreements occur, experts carefully review the reference answers, discuss the intended interpretation in detail, and continue adjusting the scoring guideline to ensure clarity and consistency.
>
> Once the scoring guidelines are finalized, the role of LLMs is to assess responses strictly according to these agreed-upon standards.
>
> **Response Q1 and Q2:** Thank you for the question. We have already discussed the performance gap between LLMs and human raters in our response to W2. Here, we provide further clarification regarding the types of biases LLMs exhibit in SAS and the extent to which these limitations can be quantified.
>
> Our analysis indicates that the most common biases arise from variation in response length and from changes in the placement of key information within an answer. In the revised version of the paper, **Appendix R** presents a detailed case study that compares model predictions with human scores across standard short-answer questions and template-free gap-filling questions.
>
> In Case 1, the responses differ in length but are correct or nearly correct. We find that both DeepSeek-V3 and Qwen2.5-14B-Instruct consistently assign higher scores to longer answers, even when shorter responses are fully aligned with human scoring criteria. We also observe that shifting the position of key content, as illustrated by the responses from students C and D, leads to substantial score variation from both models.
>
> Case 2 examines template-free gap-filling questions containing fully incorrect answers embedded within longer responses. In one example, we manually swap the concepts “dominant” and “recessive.” Qwen2.5-14B-Instruct frequently overestimates these incorrect responses and assigns inflated scores, while DeepSeek-V3 shows stronger robustness. This pattern is consistent with the quantitative results reported in Table 2.
>
> Taken together, these findings show that current LLMs remain sensitive to superficial features such as response length and content order. Although template-free gap-filling reduces these biases for stronger models, smaller models are still heavily affected. Combined with the overall performance results, these observations demonstrate that SAS-Bench is not saturated and that there remains substantial room for improvement in future LLMs.

---

> > ### Comment · Reviewer_aiMU · 2025-11-22
> >
> > Thank you for your reply, I've adjusted my score accordingly.
> >
> > I think that the motivation clarification given in your reply should be incorporated into the paper to emphasize exactly why SAS was chosen from all possible tasks that feature LLMs as judges. Looking at Appendix J again - I see that the gap exists, but it still seems somewhat insignificant, perhaps this could also be more clearly explained or reworked. I also think that the examples given in Appendix R were helpful in understanding the benchmark.
> >
> > Lastly, I think that evaluating on GPT-5 or Claude 4.5 Opus could also strengthen the results, especially if the gap between human agreement and LLM agreement remains as it is.
> >
> > Minor: In Appendix R, the text seems to be squished.

---

> > > ### Author Response · Authors · 2025-11-23
> > >
> > > Thank you for your positive feedback. We also appreciate your careful review, which helped us identify the squished text issue in **Appendix R**. We have corrected this problem by replacing the images with properly formatted versions.
> > >
> > >
> > > * Regarding the motivation clarification, we have incorporated the **Response from W1** into the Introduction to more clearly articulate why SAS was chosen among the various tasks in which LLMs serve as judges.
> > >
> > > * For **Appendix J**, we acknowledge that the previous experimental data and explanations were not sufficiently detailed or precise. To address this, we expanded the experiments by adding additional SLM baselines. *(During the verification of the Table 5 results, we found that the earlier BERT and ERNIE 3.0 results were produced under a few-shot setting that had originally been designed to align with the zero-shot experiments in Table 6. We apologize for the confusion that this caused regarding the observed gap. Because the amount of training data in this setting was very limited, the resulting performance gap appeared smaller.)* To provide a more complete analysis, we have now included SLM few-shot and full-shot experiments, LLM zero-shot experiments that exclude the step-wise annotations used in Table 6, and full-shot SFT experiments. The updated results are as follows:
> > > | **Model** | **LE** | **ASAG** | **SR** |
> > > | -------------------- | ---------------- | ---------------- | ---------------- |
> > > | **Zero-shot** | | | |
> > > | Qwen3-8B | 61.74 (1.05) | 38.58 (1.44) | 42.69 (1.72) |
> > > | Llama-8B-Instruct | 79.88 (1.81) | 37.91 (2.19) | 38.47 (1.58) |
> > > | GLM4-9B-Chat | 82.23 (2.41) | 28.39 (2.77) | 51.07 (2.11) |
> > > | Qwen2.5-32B-Instruct | 88.92 (2.73) | 69.41 (2.66) | 62.28 (2.12) |
> > > | Qwen3-32B | 89.52 (1.48) | **70.76 (1.96)** | 69.85 (2.91) |
> > > | **Few-shot** | | | |
> > > | BERT | 88.98 (0.27) | 58.23 (0.14) | 33.18 (0.25) |
> > > | SPRAG | 89.18 (0.39) | 58.18 (0.01) | 34.92 (0.21) |
> > > | M-Sim | 89.24 (0.04) | 58.26 (0.20) | 41.87 (0.19) |
> > > | ERNIE 3.0 | **93.24 (0.12)** | 58.08 (0.11) | **71.81 (0.23)** |
> > > | **Full-shot** | | | |
> > > | Qwen3-8B | 76.77 (2.14) | 37.81 (1.72) | 42.33 (2.89) |
> > > | Llama-8B-Instruct | 84.22 (2.67) | 34.62 (1.35) | 47.03 (2.48) |
> > > | GLM4-9B-Chat | 84.66 (1.91) | 30.01 (2.73) | 46.00 (1.64) |
> > > | Qwen2.5-32B-Instruct | 91.80 (1.88) | 60.24 (2.31) | 64.49 (2.07) |
> > > | Qwen3-32B | 91.88 (1.43) | 52.43 (2.56) | 55.61 (1.98) |
> > > | BERT | 89.78 (0.05) | **75.35 (0.09)** | 67.48 (0.10) |
> > > | SPRAG | 89.98 (0.04) | 75.28 (0.04) | 68.52 (0.01) |
> > > | M-Sim | 90.18 (0.03) | 74.18 (0.04) | 68.34 (0.02) |
> > > | ERNIE 3.0 | **92.61 (0.09)** | 73.49 (0.10) | **74.35 (0.07)** |
> > >
> > > Based on the updated experiments, the revised analysis in **Appendix J** leads to the following conclusions:
> > >
> > > ```
> > > We designed two sets of experiments for SLMs. The first follows a full-shot setting, in which models are trained on all available data (LE 409, ASAG 440, SR 231). The second adopts a few-shot setting with an “N-way K-shot” configuration. Each question forms an N-way task, and K is set to 2, resulting in 200 samples for LE, 42 for ASAG, and 68 for SR. This setup allows for a comparison of model performance under different data regimes. We evaluated several representative SLMs, including vanilla BERT-base, SPRAG, M-Sim, and ERNIE 3.0-base with 0.11B parameters, and compared them with LLMs using zero-shot inference and full-shot training via supervised fine-tuning (SFT). The results are summarized in Table 5.
> > >
> > > As shown in the table, several LLMs perform worse in the full-shot setting on ASAG and SR than in the zero-shot setting. A possible explanation is that increasing single-domain training data may exacerbate catastrophic forgetting on this relatively simple downstream scoring task, thereby reducing generalization ability. Overall, LLMs demonstrate advantages over SLMs in the few-shot setting, which is expected given the limited capacity of SLMs and their reliance on larger amounts of training data to achieve strong performance. However, even in the SLMs' few-shot setting, the strongest LLM in our experiments still underperforms ERNIE 3.0 on the LE and SR datasets. In the full-shot regime, the performance gap between LLMs and SLMs becomes more pronounced, with all SLMs substantially outperforming LLMs on ASAG and SR. These findings provide empirical support for our observation that LLMs do not consistently surpass SLMs on SAS tasks. While LLMs may be suitable for zero-shot scoring scenarios, their performance ceilings in supervised SAS remain an open challenge that warrants further investigation.
> > > ```
> > >
> > > * Regarding your suggestion to evaluate GPT-5 or Claude 4.5 Opus, due to current API limitations and the substantial time required for large-scale evaluation, we plan to include these models in future work.
> > >
> > > Thank you again for your thoughtful comments. We are happy to answer any further questions you may have, and we wish you a pleasant weekend.

---

### Official Review · Reviewer_H3Ty · 2025-10-31

**Soundness:** 2
**Presentation:** 3
**Contribution:** 2
**Rating:** 4
**Confidence:** 3

**Summary:**

The paper presents a new benchmark for short answer scoring/grading. The main
difference compared to existing benchmarks are that SAS-Bench contains
step-wise scoring for each answer and that errors have been categorized by
domain experts. Overall the dataset contains around 1K questions and 4K
answers. The dataset is evaluated using 16 LLMs with a focus on overall score,
step-by-step scores and error types.

**Strengths:**

S1: The detailed, step-by-step score annotation as well as the error type annotation is quite useful and a good addition to existing datasets.

S2: The introduced scores for overall, step-by-step and error are sensible and the evaluation in terms of number of LLMs quite extensive.

**Weaknesses:**

W1: The student answers seem to be mostly generate by LLMs. Some answers were generated by only six students. The distribution is not clear. How many are from students? How many of the eight generated answers per question are disregarded?

W2: The student answers are not real answers, collected by students that actually have taken the test. This means that the dataset most likely does not contain many of the patterns found in real student responses such as empty and half-completed answers.

W3: Dataset is only in Chinese. It would be more useful to have an English version as well, also to compare inter-lingual differences in terms of performance.

**Questions:**

Q1: Why are there only roughly 4 student responses per question? This seems very low.

Q2: What is the difference between step-by-step scores and errors? These seem highly related. A correct step will have no error and full score, whereas an error type will have a reduced score.

Q3: Can you add other languages and make the dataset multilingual?

Q4: From the paper it is not clear how many answers have been actually created by students and how many by LLMs. Also, having overall six student annotators is very little and the setting seems not realistic (i.e., these students are given the task to annotate and have not taken the exam)

Q5: Did you perform any evaluation on adversarial attacks on the LLM graders?

---

> ### Author Response · Authors · 2025-11-21
>
> **Response W1 and Q1:** Thank you for your comment. It is important to clarify that SAS-Bench is designed as an inference-only benchmark for evaluating whether LLMs can align with human scoring when following the scoring guideline. From this perspective, the number of samples is sufficient for reliable evaluation, even though the dataset is not intended for model training.
>
> Then, we address the concern regarding the number of real student responses from two perspectives.
>
> First, we consider the issue of score distribution. Real student responses tend to cluster in the medium to high score range. This portion of the data is also the most challenging to synthesize because it contains diverse and authentic error patterns. In contrast, LLMs can easily rewrite reference answers to create strong high-scoring responses and can also generate low-scoring answers with clear and intentional mistakes. These synthetic samples help compensate for the imbalance created by the generally higher performance of real students. To ensure that the dataset covers the full spectrum of ability levels, we use LLMs to produce both high-scoring and low-scoring responses. Appendix H, Figure 11 presents the score distributions of synthetic and real responses. The figure shows that real responses mainly fall between approximately 0.4 and 0.7, while LLM-generated responses effectively fill the high and low ends of the distribution.
>
> Second, we take annotation cost into account. Although the dataset contains 4.1 thousand responses, each response requires detailed expert annotation, including step segmentation, step scoring, and error labeling. The table in the paper reports a total of 14,573 annotated steps across subjects. As the number of samples increases, the annotation workload and associated cost grow substantially. We believe that the current dataset already provides adequate sample size and sufficient diversity in scoring patterns to support meaningful and reliable model evaluation.
>
> | | | | | | | | | | | | | | |
> | ---------- | ------------- | ------------- | ------------- | ------------- | ------------- | ------------- | ------------- | ------------- | ------------- | ------------- | ------------- | ------------- | -------- |
> | **Metric** | **Phy. (S.)** | **Phy. (M.)** | **His. (S.)** | **Geo. (S.)** | **Bio. (G.)** | **Chi. (G.)** | **Chi. (S.)** | **Math (S.)** | **Math (G.)** | **Pol. (S.)** | **Eng. (G.)** | **Che. (G.)** | **SUM.** |
> | Count | 584 | 111 | 1127 | 459 | 1626 | 402 | 1734 | 4336 | 468 | 1114 | 2357 | 255 | 14573 |
>
> **Response W2:** Thank you for the comment. As noted in our previous response, the students we were able to recruit generally perform at a medium to high level and were not working under time pressure when completing the tasks. As a result, it is difficult to obtain responses with a balanced score distribution, especially cases such as empty or partially completed answers. We therefore consider it more practical to use LLMs to synthesize such responses. LLMs can intentionally generate answers with clear errors or omissions, which are much less common in the real collected data, and thus help ensure broader coverage of the response patterns observed in real educational settings.
>
> **Response W3 Q3:** Thank you for the suggestion. We have already constructed an English version of the dataset using GPT-4o for translation, followed by human verification. Once the verification process is complete, we will release the English version as well.
>
> **Response Q2:** Thank you for the question. Step-wise scores and step-wise errors are related but capture different aspects of the evaluation.
>
> Step-wise scores reflect the number of points assigned to each step based on the scoring guideline. For instance, in a mathematics proof question, a step that states the correct formula but carries out the calculation incorrectly may receive one point, while a completely correct step may receive two points. Step-wise scores therefore measure how well the model can implicitly judge the quality of each step according to the official rubric.
>
> Step-wise error labels, in contrast, explicitly identify the underlying causes of mistakes. The model must select the correct error category from a predefined list by considering both the scoring guideline and the assigned step score. This provides a more direct and interpretable explanation of why a step is incorrect, rather than only how many points it receives.

---

> ### Author Response · Authors · 2025-11-21
>
> **Response Q4:** Thank you for the comment. As mentioned in our response to W1 Q1, Appendix H, Figure 11 presents the score distributions of both synthetic and real responses. The primary purpose of our benchmark is to evaluate whether current LLMs can accurately assess student responses across different score levels by following the scoring guidelines and referencing the questions and model answers. The goal is to measure the alignment between LLM scoring and human evaluation, rather than to exactly replicate real exam conditions.
>
> Although the recruited students were aware of the benchmark’s objectives, their task was to respond according to their actual ability. While their performance may differ slightly from what would be observed under real exam conditions, such variations fall within a reasonable margin of error and do not compromise the validity of the benchmark for evaluating LLM scoring capabilities.
>
> **Response Q5:** Thank you for the question. Our benchmark primarily focuses on evaluating LLM scoring performance under typical response conditions and does not currently consider the effects of adversarially injected phrases. For standard responses, we analyzed the impact of response length and positional bias on model performance, as reported in Appendix M. Regarding positional bias, the results demonstrate that it substantially affects scoring accuracy, suggesting that many LLMs struggle to maintain robustness against variations in response position, which likely contributes to inconsistencies across subjects.
>
> Regarding length bias, the results indicate that for both gap-filling and multiple-choice tasks, discrepancies between model predictions and human annotations vary noticeably with response length, confirming that the template-free design effectively captures length-related effects. Furthermore, deviations in step-wise scores are generally more pronounced than those in overall scores, regardless of question type. In particular, as response length increases, step-wise predictions diverge more substantially from human annotations than overall score predictions. These findings underscore the necessity of our step-wise design for effectively detecting and analyzing length bias.

---

> ### Author Response · Authors · 2025-11-28
>
> Dear Reviewer,
>
> We thank again for your contributions to the reviewing process. The responses to your concerns and the corresponding paper revision have been posted. Please let us know whether we have properly addressed your concerns. We look forward to your reply and welcome any further questions.
>
> Best regards,
>
> Authors of Paper: "SAS-Bench: A Fine-Grained Benchmark for Evaluating Short Answer Scoring with Large Language Models"

---

> > ### Comment · Reviewer_H3Ty · 2025-11-28
> >
> > Thank you for your detailed responses. I have read them, but I will keep my initial score.

---

### Official Review · Reviewer_fNhs · 2025-11-01

**Soundness:** 2
**Presentation:** 3
**Contribution:** 2
**Rating:** 4
**Confidence:** 3

**Summary:**

This paper introduces SAS-bench, a benchmark to evaluate LLM-based short answer scoring (SAS) by providing fine-grained, step-wise scoring, expert-annotated error categories, and a diverse range of question types derived from real-world subject-specific exams.

**Strengths:**

* SAS-bench aims to provide fine-grained analysis and explanations behind LLM-based SAS scoring, as well as actionable feedback. These problems are highly relevant in the EduNLP community.
* The approach to splitting answers into reasoning steps and evaluating each step for correctness and error analysis is intuitive to obtain fine-grained analysis.
* The authors release the SAS-bench publicly, containing 1030 questions from a real-world exam (China’s National College Entrance Examination(Gaokao)) with 4109 student responses.
* Auxiliary results showing in-context examples and rubrics help improve scoring performance.
* Comprehensive evaluation of 16 LLMs across different families and sizes.

**Weaknesses:**

* Since the primary contribution is a dataset, the synthetic nature of the student responses needs to be justified as well-aligned to responses from real-world test student takers. Reference responses are first generated by only 3 students, thereby lacking diversity. LLMs are then prompted to diversify and introduce errors to generate the final set of positive and negative responses. How well do LLMs perform at this synthetic task? Each LLM-based synthetic step should be evaluated for performance and error analysis. Prior work has shown prompting to be a poor simulator of students, with fine-tuning of real-world student responses required (SMART: Simulated Students Aligned with Item Response Theory for Question Difficulty Prediction, EMNLP 2025).
* Generalizability to new question banks and domains: Since the error taxonomy, a critical component to get fine-grained analysis,  involves a manual intervention for consolidation and refinement, how generalizable is this scoring approach for fine-grained analysis to Q from other exams/benchmarks? Was a completely automated LLM-based approach (with few-shot human-written examples) tried out?
* What is the human performance upper bound? No inter-annotator agreement is reported on the error and scoring metrics? Without human performance and agreement, we don’t know whether this task is well defined and how much behind/ahead LLM performance is compared to humans.
* Motivation: Breaking answers into steps seems intuitive to get fine-grained analysis. However, scoring each step and then adding all scores to arrive at the overall score needs to be justified. Usually, scoring rubrics include key intermediate results needed for different scores, or which missing steps/results would lead in a deduction of points. For example, for reasoning-based domains like math and physics, there could be multiple reasoning paths with varied steps, as well as many steps (max steps for math reported is 25). Students could also include simpler steps or combine or omit them. Won’t this make an alignment between the sum of step-scores and the overall score hard?
    * Instead of a step-score, simply using binary step correctness seems sufficient to provide fine-grained analysis as well as a useful indicator to arrive at the final score. This could also result in simplified evaluation metrics, with line 313 stating an instability in step-wise consistency evaluation. For example, why does the correct step 1 in Figure 2 get a score of 2, and the other correct steps get a score of 1? Won’t these per-step scoring rubrics be hard to design to avoid variance and ensure high agreement? For math and physics, line 474 states that step-wise labels may mislead the model.
    * Does adding step-wise evaluation help overall scoring performance (QWK)? The key aim is to improve overall scoring, since even if the fine-grained analysis is useful, if the overall scoring is poor, these models are not practically useful. Line 391 states that adding step-wise consistency introduces additional challenges to the model. Compared with SOTA overall-only scoring baselines, which omit step-wise analysis, how far ahead/behind are SAS-bench-trained models?
* What is the motivation behind the ECS metric design; does it extend a well-accepted existing metric? Does it correlate with human evaluation? When working with error distributions and frequencies, doesn’t ECS omit the ordering of steps with their errors? Including ordering seems key since an ordered list of steps with errors is being evaluated. Further, results on ECS (table 3) vary notably across different subjects and question types (line 454), showing that a different model is usually the best for different question types, and also has high variance. How would a practitioner choose the best model?

**Questions:**

* How is a step defined, especially in non-math/non-STEM contexts? Is there high agreement between human annotators for step decomposition?
* An education-specific evaluation needs to be performed. The major downstream application is the potential to provide useful feedback (line 205). Is this achieved by a human (teacher/student) evaluation?
* Line 385: observe positive correlation: what’s the exact number and correlation metric used?

**Details Of Ethics Concerns:**

IRB approval for reference responses from student annotators. Approval to include exam questions in the benchmark.

---

> ### Author Response · Authors · 2025-11-21
>
> **Response W1:** Thank you for your comment. It is important to clarify that SAS-Bench is designed as an inference-only benchmark for evaluating whether LLMs can align with human scoring when following the scoring guideline. From this perspective, the number of samples is sufficient for reliable evaluation, even though the dataset is not intended for model training.
>
> Regarding the concern about the number of real student responses, our rationale is as follows. First, there is an issue of score distribution. We observed that real student responses tend to concentrate around medium to high score levels. To ensure that the dataset spans a broad range of performance levels, we used LLMs to generate both high-scoring and low-scoring responses. Appendix H, Figure 11 presents the score distributions of both synthetic and real responses, and it can be seen that real responses mainly fall within the approximate range of 0.4 to 0.7.
>
> Second, there is the consideration of annotation cost. Although the dataset contains 4.1k responses, each response requires expert annotation for step segmentation, step scoring, and error labeling. The table in the paper reports a total of 14,573 annotated steps across subjects. The required effort and cost increase substantially as the number of samples grows. We believe that the current dataset already provides sufficient sample size and score diversity to support reliable model evaluation.
>
> With respect to the concern about whether synthetic responses can adequately simulate real student behavior, we acknowledge that LLMs are not naturally suited to imitating students or producing realistic erroneous responses. Our synthesis design explicitly addresses this challenge, and as described in the paper, LLMs tend to be more effective at producing either fully correct responses or responses with obvious mistakes. In practice, and as reflected in the score distribution in Figure 11, LLMs are able to rewrite reference answers to produce strong high-scoring responses, and they can also generate low-scoring responses with clear errors. These synthetic samples help mitigate the uneven score distribution that results from real students generally performing at higher levels. Furthermore, we conducted an additional round of manual review to retain only high-quality synthetic responses.
>
> | | | | | | | | | | | | | | |
> | ---------- | ------------- | ------------- | ------------- | ------------- | ------------- | ------------- | ------------- | ------------- | ------------- | ------------- | ------------- | ------------- | -------- |
> | **Metric** | **Phy. (S.)** | **Phy. (M.)** | **His. (S.)** | **Geo. (S.)** | **Bio. (G.)** | **Chi. (G.)** | **Chi. (S.)** | **Math (S.)** | **Math (G.)** | **Pol. (S.)** | **Eng. (G.)** | **Che. (G.)** | **SUM.** |
> | Count | 584 | 111 | 1127 | 459 | 1626 | 402 | 1734 | 4336 | 468 | 1114 | 2357 | 255 | 14573 |
>
> **Response W2:** Thank you for the comment. In Section 3.2 on Error Cause Construction, we state that “we propose constructing a structured list of potential error causes for each subject and question type. By constraining LLMs to select error causes from this predefined list when explaining their scoring decisions, we enable a more systematic and quantifiable evaluation of model explainability. Specifically, for each subject and question type, we adopt a few-shot prompting strategy using representative questions, reference answers, and filtered responses as input to GPT-4o, prompting it to generate 50 sets of error cause descriptions. Human annotators then consolidate and refine these descriptions, merging similar entries and simplifying the language to produce a comprehensive set of error causes that captures all major categories of mistakes.”
>
> This method combines few-shot examples with LLM-assisted generation, followed by human verification. In practical applications, the entire process could be fully automated through an LLM-based pipeline. However, for the benchmark presented in our work, we intentionally retained human verification to ensure rigor and to guarantee the quality and reliability of the resulting error taxonomy.

---

> ### Author Response · Authors · 2025-11-21
>
> **Response W3:** Thank you for the comment. We agree that the absence of human performance estimates makes it difficult to objectively evaluate LLM performance on this task. To address this, we have included human performance results in Appendix P. For each subject, two human experts independently scored the responses following the scoring guidelines, and the final score was calculated as the average of their annotations. The results show that human performance exceeds 90% on QWK, CCS, and ECS. Although consistency on short-answer questions is slightly lower than for other question types, it still remains above 85% on CCS. These findings indicate that there is a substantial gap between LLMs and human evaluators in terms of scoring consistency.
>
> | | | | | | | | | | | | | | |
> | ---------- | ------------- | ------------- | ------------- | ------------- | ------------- | ------------- | ------------- | ------------- | ------------- | ------------- | ------------- | ------------- | -------- |
> | **Metric** | **Phy. (S.)** | **Phy. (M.)** | **His. (S.)** | **Geo. (S.)** | **Bio. (G.)** | **Chi. (G.)** | **Chi. (S.)** | **Math (S.)** | **Math (G.)** | **Pol. (S.)** | **Eng. (G.)** | **Che. (G.)** | **Avg.** |
> | QWK | 87.72 | 96.59 | 88.63 | 98.87 | 97.45 | 95.64 | 94.73 | 95.84 | 93.81 | 90.64 | 92.19 | 92.86 | 93.75 |
> | CCS | 85.74 | 95.39 | 88.02 | 97.08 | 96.57 | 92.39 | 92.06 | 94.75 | 92.37 | 90.38 | 86.81 | 95.52 | 92.26 |
> | ECS | 86.77 | 89.45 | 93.81 | 93.79 | 92.83 | 93.14 | 94.30 | 98.10 | 90.73 | 90.67 | 87.29 | 96.40 | 92.27 |
>
> **Response W4:** Thank you for your comment. We address your points in two parts.
>
> (1) First, regarding the granularity of step segmentation and the definition of step scores. The different point values assigned to steps in Figure 2 directly follow the official scoring guidelines. For example, in a ten-point mathematics proof question, the guideline specifies that a correct definition of a key formula is worth one point and a correct calculation is worth two points. The model assigns scores according to these rules. In cases where students merge, simplify, or omit steps, the human annotators have already consolidated the original reasoning into coherent step units during the annotation process. As a result, the step boundaries are not affected by the structure of the student’s initial solution.
>
> With respect to the choice of step score values, we consider a binary correct or incorrect label insufficient for subjects that require nuanced argumentation, such as political science or history. The quality of reasoning in such subjects cannot be meaningfully represented by a binary label, and such a setup would deviate from real grading practice. For science and mathematics questions that involve many steps, the scoring rubric usually prescribes a fixed range of point values for each type of reasoning step, commonly between zero and three. As shown in the earlier scoring guideline example, the rubric provides clear criteria that determine the score for each step type, which supports consistent annotation.
>
> The instability noted in line 313 pertains primarily to ECS, which evaluates error-cause consistency. This metric requires the model to produce fine-grained and interpretable reasoning, which makes the evaluation more demanding and inherently less stable. In contrast, CCS remains relatively stable. In the main experiments, we also report standard errors. For example, the average CCS standard deviation of DeepSeek V3 remains within two points.
>
> (2) Second, regarding whether step-wise evaluation improves overall scoring performance, we computed the Pearson correlation between CCS and QWK across subjects for different models. Most models exhibit a positive correlation, and more than half show a strong positive correlation. As noted in line 391, a weak positive correlation indicates that a model may perform well in overall score consistency while being less consistent at the step level. This does not imply instability in QWK.
>
> | | | | |
> | ---------------------- | --------- | --- | ----- |
> | Model | Pearson R | | $$Δ$$ |
> | Deepseek-R1 | 0.840 | | -1.31 |
> | QwQ-32B | 0.620 | | 2.78 |
> | TinyR1-32B-Preview | 0.899 | | 2.45 |
> | Qwen3-32B | 0.485 | | 2.91 |
> | Qwen3-8B | 0.222 | | -2.67 |
> | MiMo-7B-RL | 0.560 | | 2.12 |
> | Deepseek-Prover-V2-7B | 0.879 | | -2.98 |
> | DeepSeek-R1-Distill-7B | -0.124 | | 3.27 |
> | Deepseek-V3 | 0.633 | | 1.85 |
> | GPT 4o-mini-20240718 | 0.272 | | -1.23 |
> | Llama3.3-70B-Instruct | 0.930 | | 2.46 |
> | Mixtral 8×7B-Instruct | 0.740 | | -2.89 |
> | Qwen2.5-32B-Instruct | 0.493 | | 2.55 |
> | Qwen2.5-14B-Instruct | 0.688 | | -2.77 |
> | GLM4-9B-Chat | 0.451 | | 2.33 |
> | Llama3-8B-Instruct | 0.313 | | -3.99 |
>
> We also compared QWK with and without step-wise scoring; the $\Delta$ values, showing minor increases and decreases, indicate that step-wise evaluation has minimal impact on overall performance.

---

> ### Author Response · Authors · 2025-11-21
>
> **Response W5:** The motivation behind ECS is to evaluate the model’s ability to identify and analyze the causes of errors in student responses, thereby enhancing the interpretability of its scoring behavior. Structurally, ECS is computed using Spearman correlation as a sequence similarity measure, where the sequence consists of the frequency counts of different error types.
>
> Regarding the role of step order, we did consider incorporating the sequential structure of errors during the design of ECS. However, assessing error-cause consistency at the level of individual samples introduces substantial instability. This instability arises primarily from the sparsity of error types at the step level, where each step typically contains only a small subset of possible errors. Under such circumstances, incorporating both order and frequency would introduce considerable noise and undermine the reliability of the evaluation. To improve stability, we instead aggregate samples into high, medium, and low score groups. Increasing the number of samples within each group yields more stable estimates. However, once samples are aggregated, it is no longer possible to derive a meaningful error sequence order, so we retain error frequency as the primary factor.
>
> Even with these adjustments, model performance on ECS remains relatively weak. For this reason, we believe that focusing on error-frequency distributions is sufficient at this stage for assessing the model’s ability to provide interpretable error-cause analyses.
>
> **Response Q1:** Thanks for your comment. Regarding the definition of a step, we provide detailed explanations for different question types in Appendix O. Before annotation, subject experts discussed and agreed on the guidelines for step segmentation to ensure consistency across annotators. As stated in the appendix:
> “To enable fine-grained evaluation, we develop step division rules tailored to different question types. Representative examples of each case are provided in Figure~\ref{f11}. Overall, the division of response steps falls into three categories.”
>
> **Case 1 (Reasoning and Calculation Based).**
> This category includes subjects such as Mathematics (S.) and Physics (S.). These questions require systematic reasoning and sequential calculations. Responses are segmented into steps based on key reasoning units.
>
> **Case 2 (Argument Based).**
> This category includes subjects such as Geography (S.), parts of Politics (S.), and Chinese (S.). These questions typically require argumentative or discursive responses, which are divided into steps according to distinct claims or supporting pieces of evidence.
>
> **Case 3 (Pre-set Question Structure).**
> This category includes subjects such as Mathematics (G.), Politics (S.), Chinese (G.), Chinese (S.), English (G.), Physics (G.), Biology (G.), History (S.), and Chemistry (G.). In these cases, the question itself defines a clear multi-part structure, and responses are segmented directly according to the sequence of sub-questions.
>
> We also note that step segmentation follows a standardized annotation protocol, and experts reached agreement on the segmentation criteria before conducting the annotations.
>
> **Response Q2:** Thank you for the suggestion. As noted in line 205, the current use of LLM-based error cause consistency feedback is primarily intended to help humans assess the reliability of model-generated scores. Its effectiveness as direct educational feedback is difficult to quantify at this stage. To better understand the present capabilities of LLM scoring, we compare model performance with human scoring consistency, which provides a quantitative measure of the gap between LLMs and human evaluators. The human scoring results are reported in our **response to W3**.
>
> **Response Q3:** Thank you for the question. We use the Pearson correlation coefficient to measure the relationship between CCS and QWK. The detailed results and conclusions are provided in our response to Question W4.

---

> ### Author Response · Authors · 2025-11-28
>
> Dear Reviewer,
>
> We thank again for your contributions to the reviewing process. The responses to your concerns and the corresponding paper revision have been posted. Please let us know whether we have properly addressed your concerns. We look forward to your reply and welcome any further questions.
>
> Best regards,
>
> Authors of Paper: "SAS-Bench: A Fine-Grained Benchmark for Evaluating Short Answer Scoring with Large Language Models"

---

### Meta-Review · Area_Chair_E9gV · 2026-01-07

**Summary:**

Overall, reviewers agree that SAS-Bench tackles an important problem in short-answer scoring and that step-wise scoring with expert error-cause annotations is the paper’s main strength. The suggested decision is shaped by recurring concerns about the realism of LLM-synthesized student responses, the subjectivity of step-level annotations, and limited evidence of generalizability beyond a single exam and language. Several reviewers also questioned whether the proposed metrics and step-wise formulation are sufficiently validated or practically necessary. While the rebuttal substantially improved clarity and added human upper-bound results, some foundational concerns remain only partially resolved.

**Reviewer Concerns:**

Reviewer fNhs: The rebuttal addressed human upper-bound performance, clarified step definitions across subjects, and explained the ECS design, but concerns about the realism of synthetic responses, lack of explicit inter-annotator agreement statistics, limited evidence of generalizability, and absence of an education-specific usefulness evaluation remain outstanding.

Reviewer H3Ty: The rebuttal clarified dataset size, the role of synthetic answers, the distinction between step-wise scores and errors, and plans for an English version, but concerns about realism of student answers and the lack of adversarial robustness evaluation were not fully resolved.

Reviewer aiMU: The rebuttal effectively clarified the motivation, addressed saturation concerns by contrasting human and model agreement, and provided concrete bias analyses, with the main remaining issue being the absence of formal inter-annotator agreement measures.

Reviewer 5Cgo: The rebuttal addressed variance reporting and partially addressed language generalization, but concerns about differentiation from prior SAS benchmarks, explicit inter-annotator agreement reporting, and empirical analysis of synthetic-data bias remain unresolved.

**Reviewer Scores:**

Reviewer fNhs (4) did not indicate a score change, and given that several of their core methodological concerns were only partially addressed, it is likely the rebuttal did not fully resolve all of their questions.

Reviewer H3Ty (4) explicitly stated that they would keep their initial score after reading the rebuttal.

Reviewer aiMU (4) explicitly stated that they adjusted their score after the rebuttal, indicating that their main concerns were largely addressed.

Reviewer 5Cgo (6) did not comment on score changes, and since several of their key concerns remain unresolved, the rebuttal does not appear to have fully addressed all of their questions.

---

### Decision · Program_Chairs · 2026-01-26

Reject